# The Impact of Positional Encoding on Length Generalization in Transformers

**Amirhossein Kazemnejad**[1], **Inkit Padhi**[2]
**Karthikeyan Natesan Ramamurthy**[2], **Payel Das**[2], **Siva Reddy**[1,3,4]
[1]Mila, McGill University; [2]IBM Research;
[3]Facebook CIFAR AI Chair; [4]ServiceNow Research
{amirhossein.kazemnejad,siva.reddy}@mila.quebec
inkpad@ibm.com, {knatesa,daspa}@us.ibm.com

## Abstract

Length generalization, the ability to generalize from small training context sizes to larger ones, is a critical challenge in the development of Transformer-based language models. Positional encoding (PE) has been identified as a major factor influencing length generalization, but the exact impact of different PE schemes on extrapolation in downstream tasks remains unclear. In this paper, we conduct a systematic empirical study comparing the length generalization performance of decoder-only Transformers with five different position encoding approaches including Absolute Position Embedding (APE), T5's Relative PE, ALiBi, and Rotary, in addition to Transformers without positional encoding (NoPE). Our evaluation encompasses a battery of reasoning and mathematical tasks. Our findings reveal that the most commonly used positional encoding methods, such as ALiBi, Rotary, and APE, are not well suited for length generalization in downstream tasks. More importantly, NoPE outperforms other explicit positional encoding methods while requiring no additional computation. We theoretically demonstrate that NoPE can represent both absolute and relative PEs, but when trained with SGD, it mostly resembles T5's Relative PE attention patterns. Finally, we find that scratchpad is not always helpful to solve length generalization and its format highly impacts the model's performance. Overall, our work suggests that explicit position encodings are not essential for decoder-only Transformers to generalize well to longer sequences.

## 1 Introduction

The ability to generalize from smaller training context sizes to larger ones, commonly known as length generalization, is a major challenge for Transformer-based language models (Vaswani et al., 2017; Deletang et al., 2023; Zhang et al., 2023). Even with larger Transformers, this issue persists (Brown et al., 2020; Furrer et al., 2020). With larger context sizes, a model can benefit from more in-context-learning examples, higher numbers of reasoning steps, or longer text generation. However, training a Transformer with a larger context size can be excessively slow and memory-intensive. This is even more pronounced in the recent paradigm of model finetuning on instruction-following datasets (Wei et al., 2022a; Chung et al., 2022; Ouyang et al., 2022). It is not only infeasible to train the model on all possible context lengths, but also the number of training examples drops dramatically as the sequence length increases requiring the model to generalize from finite and shorter-length training examples. In this work, we focus on the effect of *positional encoding* on length generalization in the "**decoder-only**" Transformers on various tasks trained from scratch. Figure 1 summarizes our finding that using no positional encoding is better than using explicit positional encodings.

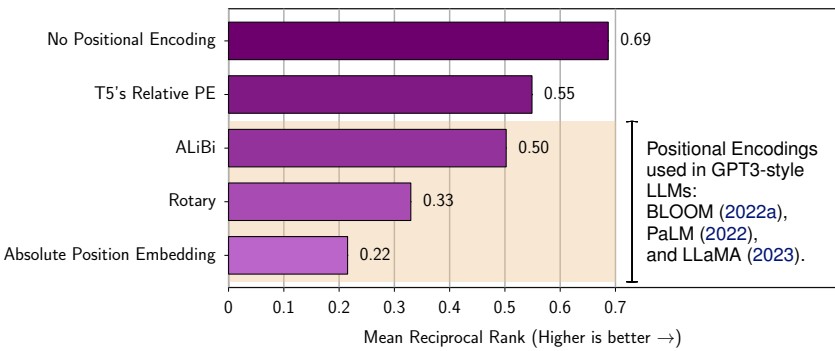

Figure 1: No positional encoding (NoPE) outperforms all other positional encodings at length generalization of decoder-only Transformers (GPT-style) trained from scratch and evaluated on a battery of reasoning-like downstream tasks. This figure shows aggregate ranking of positional encoding methods across 10 tasks.

Positional encoding (PE) seems to be a major factor in the length generalization of Transformers as the model has to systematically encode tokens in *all* possible positions. The original Transformer architecture (Vaswani et al., 2017) used non-parametric periodic functions to represent *absolute position embeddings* (APE) in a systematic manner, but further studies have shown that these functions are inadequate for length generalization (Ontanon et al., 2022). The prevailing belief is that relative PEs (Shaw et al., 2018; Raffel et al., 2020) are more effective in length generalization than APE variants (Ontanon et al., 2022; Csordás et al., 2021). However, Press et al. (2022) has shown that even Transformers with relative PEs, such as Rotary (Su et al., 2021), can be poor at length generalization. But the evaluation of PEs often relies on language modeling perplexity as a key metric (Haviv et al., 2022; Press et al., 2022) which does not always align with the performance on downstream tasks (Tay et al., 2022). This raises important questions: what exactly is the influence of positional encoding on length generalization at *downstream tasks*? Moreover, early empirical evidence shows that decoder-only Transformers without explicit position information (Tsai et al., 2019; Haviv et al., 2022) can perform as well as existing PEs in in-distribution settings, but its effects on length generalization and downstream performance are unclear.

Recently, asking models to emit intermediate computation steps into a scratchpad, also referred to as *chain-of-thought*, has been adopted to improve the length extrapolation in Transformers (Nye et al., 2021; Wei et al., 2022b). These techniques are architecture-independent and can be used with any PE method. However, it remains an open question whether these techniques, at least in regard to length generalization, render the choice of PE irrelevant, especially given that model performance is highly sensitive to the scratchpad format (Bueno et al., 2022; Akyurek and Akyurek, 2022).

To answer these questions, we conduct a systematic study on the length generalization of decoder-only Transformers, popularized by the GPT-family of models (Radford et al., 2019), with the most commonly used positional encoding schemes, both with and without scratchpad. Specifically, we evaluate APE (Vaswani et al., 2017), T5's Relative PE (Raffel et al., 2020), ALiBi (Press et al., 2022), Rotary (Su et al., 2021) and no positional encoding (NoPE) on a battery of reasoning and mathematical tasks. Our results show that:

- Most commonly used positional encoding methods, including ALiBi, Rotary, and APE, are ill-suited for length generalization in downstream tasks and are outperformed by T5's Relative PE.
- Transformers without positional encoding (NoPE) outperform all explicit positional encoding schemes. They achieve this without computing additional terms in the attention mechanism (in contrast to explicit PEs).
- We show that NoPE is theoretically capable of representing both absolute and relative PEs. But empirically, it is closer to the relative encoding scheme similar to T5's Relative PE.
- Scratchpad is not always helpful for length generalization and its format highly impacts the performance. The attention distributions reveal that NoPE and T5's Relative PE encourage attending to both long and short-range positions, ALiBi to recent positions, and Rotary and APE to no particular positions.

## 2    Background: Positional Encoding in Transformers

Transformers, in contrast to sequential models such as RNNs, are parallel architectures that employ positional encoding to help encode word order. The most common choices for positional encoding are either *absolute*, where each absolute position (e.g. 1, 2, 3, ...) is directly represented, or *relative*, where the distance between tokens is used as positional information. In this section, we briefly review the popular encoding methods used in Transformers (Refer to Appendix B for more formal details).

*Absolute Position Embedding* (APE), embeds each absolute position $i$ into position vector $\boldsymbol{p}_i$ and adds word embeddings to their corresponding $\boldsymbol{p}_i$ before feeding them to the model. The non-parametric variant of APE uses periodic functions such as sine and cosine to generate embeddings for any position $i$ (Vaswani et al., 2017). On the other hand, a learned version of APE, used in GPT3 (Brown et al., 2020) and OPT (Zhang et al., 2022), trains the position embeddings along with the model parameters, and it cannot generate a position embedding for unseen positions, so the context window is set to a fixed length.

*T5's Relative bias*, first maps the relative distance $(i - j)$ between tokens at positions $i$ and $j$ to a scalar bias value $b = f(i - j)$, where $f$ is a lookup table. The relative bias $b$ (learned during training) then is added to the dot product of the query and key in the self-attention mechanism. The lookup table maps distances larger than a threshold to the same parameter to enable generalization to unseen distances.

*Rotary*, used in PaLM (Chowdhery et al., 2022) and LLaMA (Touvron et al., 2023), rotates the query and key representations with an angle proportional to their absolute positions before applying the dot product attention. As a result of this rotation, the attention dot product will only depend on the relative distance between tokens, effectively making it a relative positional encoding (Su et al., 2021).

*ALiBi*, used in BLOOM (Scao et al., 2022a), is similar to T5's Relative Bias but instead subtracts a scalar bias from the attention score. This bias grows linearly with the distance between the query and key tokens. This, in effect, creates a preference toward recent tokens (recency bias).

Note that encoder-only Transformers, such as BERT, become bag-of-words models in the absence of positional encoding. However, decoder-only Transformers with causal attention mask are not permutation invariant and can model sequences even without explicit position information (Tsai et al., 2019). But it is unclear if these models encode position information implicitly or generalize to unseen lengths. We demystify this in Section 5.

## 3    Model Evaluation

**Length Generalization Setup**    Following Anil et al. (2022), we focus on algorithmic tasks such as copying, addition, etc. For each task, we train on a finite number of examples of up to a certain length and test them on both seen and unseen lengths at inference. We present these problems as sequence-to-sequence tasks, where the input sequence is the problem instance and the output sequence is the solution. Formally, let $\mathcal{D} = \{(\boldsymbol{x}_i, \boldsymbol{y}_i)\}$ denote a dataset of such task where $\boldsymbol{x}_i$ is the input and $\boldsymbol{y}_i$ is the output sequence. For each task a function $\lambda : \mathcal{D} \rightarrow \mathbb{N}$ can be defined that returns the length bucket of a task instance $d \in \mathcal{D}$. This can be the number of tokens or any general notion of length/depth of reasoning. Using this function and a threshold $L$, we employ samples where $\lambda \leq L$ for learning the task and samples where $\lambda > L$ for evaluating generalization. The performance on each instance is reported as the exact-match accuracy of its answer with the ground truth.

**Architecture**    We use a conventional decoder-only Transformer architecture as a base for all experiments and consider different approaches for encoding positions: **Absolute Position Embedding (APE)**, **ALiBi**, **Rotary** and **T5's Relative Bias**. We also consider removing the positional encoding (**NoPE**) to better understand its role in length generalization. Note that we use APE with sinusoidal functions (Vaswani et al., 2017) as the learnable variant cannot produce embeddings for unseen positions. Given the absence of publicly available Transformer-based LM trained with aforementioned PEs on the same pretraining data, we opt to train our models from scratch for each task on its training data with the autoregressive language modeling objective $\log p_\theta(\boldsymbol{y}|\boldsymbol{x}) = \sum_{t=1}^{T} \log p_\theta(y_t|\boldsymbol{x}, \boldsymbol{y}_{1:t-1})$. We use the same hyperparameters for all PEs and employ the "`base`" model size configuration, popular in HuggingFace library (Wolf et al., 2020), resulting in $\sim$107M trainable weights (List of all hyperparameters in Appendix D.2).

Table 1: Examples of the input and output of the tasks.

| Task | Input Example | Output Example |
|---|---|---|
| **Primitive Tasks** | | |
| Copy | `Copy the following words:  <w1> <w2> <w3> <w4> <w5>` | `<w1> <w2> <w3> <w4> <w5>` |
| Reverse | `Reverse the following words:  <w1> <w2> <w3> <w4> <w5>` | `<w5> <w4> <w3> <w2> <w1>` |
| **Mathematical and Algroithmic Tasks** | | |
| Addition | `Compute:  5 3 7 2 6 + 1 9 1 7 ?` | `The answer is 5 5 6 4 3.` |
| Polynomial Eval. | `Evaluate x = 3 in ( 3 x ** 0 + 1 x ** 1 + 1 x ** 2 ) % 10 ?` | `The answer is 5.` |
| Sorting | `Sort the following numbers:  3 1 4 1 5 ?` | `The answer is 1 1 3 4 5.` |
| Summation | `Compute:  ( 1 + 2 + 3 + 4 + 7 ) % 10 ?` | `The answer is 7.` |
| Parity | `Is the number of 1's even in [ 1 0 0 1 1] ?` | `The answer is No.` |
| LEGO | `If a = -1; b = -a; c = +b; d = +c.  Then what is c?` | `The answer is +1.` |
| **Classical Length Generalization Datasets** | | |
| SCAN | `jump twice and run left` | `JUMP JUMP TURN_LEFT RUN` |
| PCFG | `shift prepend K10 R1 K12 , E12 F16` | `F16 K10 R1 K12 E12` |

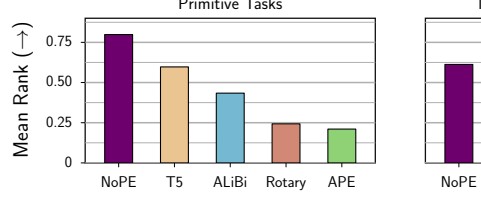 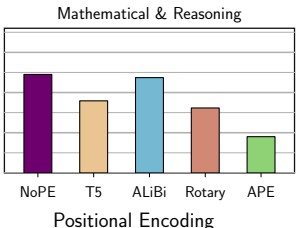 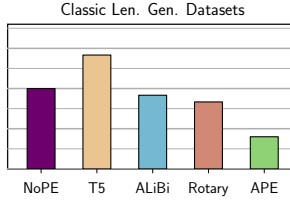

Figure 2: Aggregate ranking of positional encoding methods on length extrapolation across three different groups of tasks. No PE and T5's Relative Bias outperform other encoding methods in these categories.

**Tasks** Our study of length generalization is concentrated on downstream tasks. Particularly, we evaluate the models on three categories (Table 1) of synthetic tasks that have been widely used in the literature to investigate length generalization: (1) Primitive tasks such as Copying and Reversing (Ontanon et al., 2022), (2) Mathematical and reasoning tasks such as Addition (Nye et al., 2021), Polynomial Evaluation, Sorting, Summation (Saxton et al., 2019), Parity (Anil et al., 2022), LEGO (Zhang et al., 2023) and (3) Classical length generalization datasets such as SCAN (Lake and Baroni, 2018) and PCFG (Hupkes et al., 2020). These tasks provide us with complete control over the train-test distribution, while also requiring reasoning and compositionality skills, which serve as fundamental building blocks for more complex tasks. For the first two categories, we generate the corresponding datasets. Specifically, we first sample the length of the task instance from the uniform distribution $\mathcal{U}(1, L)$, and then, according to the task's generative process, we sample the input and output sequences. For the test set, we follow the same procedure but sample length from $\mathcal{U}(1, 2L)$ to include both seen and unseen lengths. Throughout the paper, unless otherwise stated, we use $L = 20$. For the third category of tasks, we use length generalization splits from the corresponding datasets. Table 1 provides an example of each task (More examples in Appendix D.1).

We report the results of our empirical evaluation over ten tasks and three seeds per dataset-PE pair.

## 4 What Is The Effect of Positional Encoding?

In this section we provide comparative results of positional encodings at length generalization. To provide a holistic view, following Liang et al. (2022), we report the mean ranking of various models in Figures 1 and 2 when compared against each other for all tasks and scenarios. Furthermore, we showcase the accuracy of models evaluated on examples of various lengths in Figure 3. (Detailed results for each task and scenario can be found in Appendix E).

First, we observe that in most tasks, models achieve a perfect or near-perfect accuracy (Figure 3) on the I.I.D. lengths, which indicates that models have no problem fitting to the training data. However, the differences among positional encoding methods become more apparent when we evaluate on lengths that are larger than seen during training. In most extrapolation scenarios, T5's Relative Bias

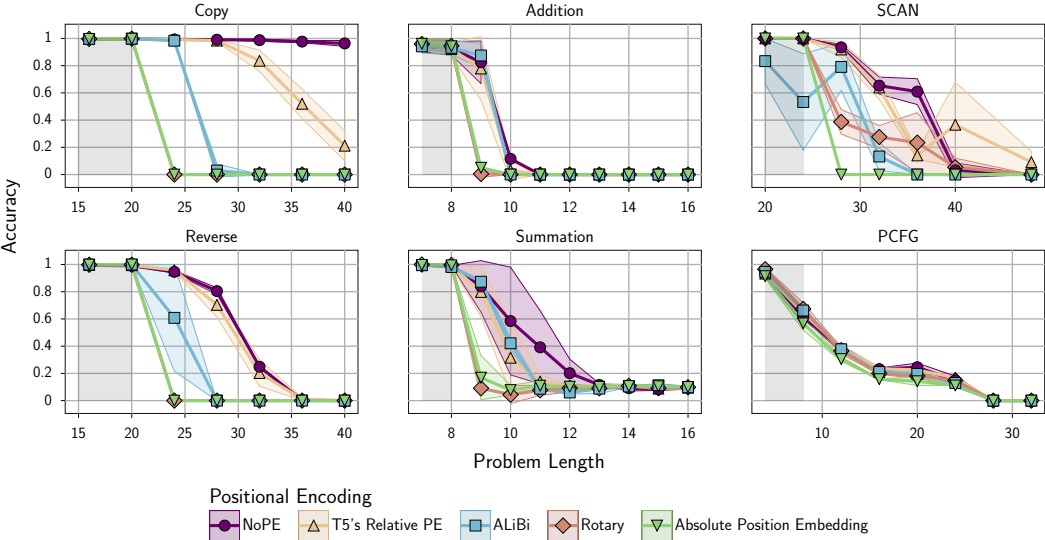

Figure 3: Showcasing the generalization behavior of different positional encodings on 6 datasets. The shaded area represents evaluation examples with I.I.D. lengths (i.e. seen during training). Since all models perform perfectly, or close to it, on the I.I.D. lengths (measured on unseen examples), for improved readability, we only show a subset of them in the figure. Refer to Appendix E for more detailed plots.

outperforms other explicit positional encodings. ALiBi positions itself in the middle of the pack, while APE and Rotary show poor generalization performance.

Although Rotary is often considered a relative encoding method (Ontanon et al., 2022), our results show that it performs more similarly to APE than to other relative schemes. Moreover, ALiBi, despite its promise for length generalization, underperforms with respect to T5's Relative Bias in most cases. This result aligns with Taylor et al. (2022) who found no significant improvement from ALiBi.

Surprisingly, the NoPE model, which is just a decoder-only Transformer without any positional encoding, performs on par with or even better than the best-performing explicit PE, T5's Relative Bias. NoPE achieves the same level of generalization without *any computational overhead* since it does not compute any additional term in the attention mechanism. This property has a direct impact on the runtime and memory footprint of the model. For instance, Press et al. (2022) reported that the additional computation incurred by T5's Relative Bias can make the training and inference time of the model almost two times slower than the Transformer with APE.

## 5 How Does NoPE Represent Positions?

The surprising performance of NoPE model suggests that it capture useful positional information that can also generalize. But, how it does so is the primary question. In the next two sections, we provide theoretical and empirical analysis towards answering this question.

### 5.1 NoPE can theoretically represent both absolute and relative PEs

Let $f_\theta$ be a NoPE decoder-only Transformer model, where $\theta$ denotes the model parameters. $f_\theta$ processes the input sequence $\boldsymbol{x} = [\texttt{<bos>}, x_1, \ldots, x_T]$ by applying a series of layers. Note that since $f_\theta$ does not have any PE, the input $\boldsymbol{x}$ is not augmented with positional information (e.g. $[1, 2, \ldots, T]$). Each layer $l$, consisting of self-attention heads and a feed-forward sub-layer, reads the previous hidden state $\boldsymbol{H}^{(l-1)}$ and produces the hidden state at layer $l$: $\boldsymbol{H}^l$. Each head is parameterized by a query $\boldsymbol{W}_Q$, key $\boldsymbol{W}_K$, value $\boldsymbol{W}_V$, and output $\boldsymbol{W}_O$ matrices, where $\boldsymbol{W}_Q, \boldsymbol{W}_K, \boldsymbol{W}_V \in \mathbb{R}^{h \times d}$ and $\boldsymbol{W}_O \in \mathbb{R}^{d \times h}$. $d$ and $h$ are the model's hidden state size and attention dimension, respectively. $\boldsymbol{W}_1, \boldsymbol{W}_2 \in \mathbb{R}^{d \times k.d}$ are the weight matrices of the feed-forward sub-layer.

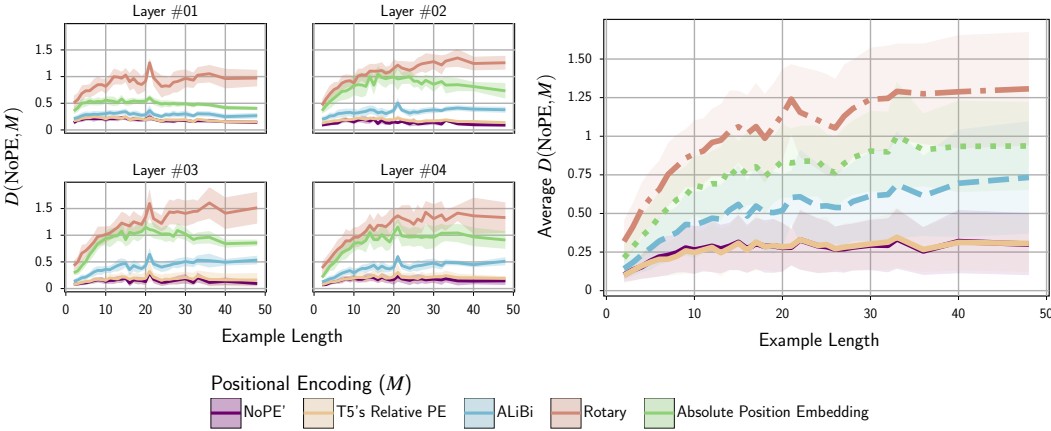

Figure 4: Distance of NoPE attention patterns with other positional encoding schemes measured across instances of SCAN dataset. The left figure shows the distance per layer, and the right figure shows the average distance across all layers. NoPE' denotes NoPE trained with a different seed.

**Theorem 1** (**Absolute Encoding**). *Let $\boldsymbol{x}$ be an input sequence of length $T + 1$ to the model. Then, the first layer of $f_\theta$ can recover absolute positions $[1, \ldots, T + 1]$ in the hidden state $\boldsymbol{H}^{(1)}$. That is, there exist $\boldsymbol{W}_Q, \boldsymbol{W}_K, \boldsymbol{W}_V, \boldsymbol{W}_O, \boldsymbol{W}_1,$ and $\boldsymbol{W}_2$ such that the self-attention and feedforward operations in the first layer compute absolute positions and write it to the next hidden state.*

We refer to Appendix C.1 for the complete proof of Theorem 1. This theorem shows that stochastic gradient descent (SGD) can potentially learn to recover absolute positions in NoPE Transformers. Next, we demonstrate how relative PE can be implemented in subsequent layers:

**Theorem 2** (**Relative Encoding**). *Suppose that the hidden state $\boldsymbol{H}^{(1)}$ contains absolute positional information, as stated in Theorem 1, and assume that it is not overwritten by any subsequent layers. Then, the self-attention in all subsequent layers can implement a relative positional encoding: there exists a parameterization of $f_\theta$ such that, for $l \geq 2$, the attention dot product between query $\boldsymbol{q}_n$ and key $\boldsymbol{k}_m$ at positions $n$ and $m$ can be expressed as:*

$$\langle \boldsymbol{q}_n, \boldsymbol{k}_m \rangle = f_{\mathrm{cnt}}(\boldsymbol{q}, \boldsymbol{k}) + f_{\mathrm{rel}}(n - m) \tag{1}$$

*where $f_{\mathrm{cnt}}$ is a function of their content, and $f_{\mathrm{rel}}$ is a function of their relative distance.*

Appendix C.2 provides the complete proof of Theorem 2. Our theoretical results suggest that SGD can choose between relative and absolute encoding in NoPE Transformers. But, what mechanism SGD learns in practice is not clear. We next investigate this question empirically.

## 5.2 NoPE learns to use relative PE in practice

In order to explore the mechanisms that NoPE employs in practice, we conduct a quantitative analysis by comparing its attention pattern to models trained with different positional encoding techniques. The hypothesis is that if NoPE utilizes a similar algorithm to other PEs, then the attention patterns of these models should be quite similar.

To this end, we feed the same input to both models and, at layer $l$, we compute the minimum distance between the attention distribution of any heads in the first model and any head in the second model. Formally, let $P_t = p(\boldsymbol{k}|\boldsymbol{q}_t)$ be a probability distribution produced by a causal self-attention head for query at position $t$, over the keys $\boldsymbol{k} \in [\boldsymbol{k}_1, \ldots \boldsymbol{k}_t]$ in a given transformer layer. Over a sequence of length $T$, we define the similarity between two heads P and Q as $D_{\mathrm{AT}}(P, Q) = \frac{1}{T} \sum_{t=1}^{T} D_{\mathrm{JSD}}(P_t || Q_t)$ which averages the Jensen–Shannon divergence (JSD) between the two heads over all positions. For the distance of two models $A$ and $B$ at layer $l$, we take the minimum distance

between all pairs of attention heads in the corresponding layer:

$$D^{(l)}(A, B) = \min_{(P,Q) \in A_l \times B_l} D_{AT}(P, Q) \tag{2}$$

where $A_l$ and $B_l$ are the attention heads in layer $l$ of models $A$ and $B$ respectively. We empirically measure the distance between NoPE and other positional encoding schemes after training. Specifically, we sample examples from each length bucket and feed them (the concatenation gold input and output) to compute the attention maps and the distance using Equation (2). We also consider the distance between different seeds of NoPE as a baseline. Figure 4 shows the distance per layer for the first four layers. (later layers show similar trends Figure F.7). We find that NoPE's attention patterns are most similar to that of T5's Relative PE, and least similar to APE and Rotary. The same trend can be observed across all layers and length buckets, and even when averaged across all layers. These results potentially suggest that a Transformer model without positional encoding, trained with stochastic gradient descent learns to represent positions in a way similar to T5's Relative PE, which is a relative positional encoding scheme.

## 6  Does Scratchpad Render The Choice of Positional Encoding Irrelevant?

In scratchpad/CoT prompting, the model generates intermediate computations required to reach the final answer as explicit parts of the output. Such mechanisms, in effect, provide a direct decomposition and storage for intermediate values, which has been shown to improve the length generalization of Transformers even at small scale (Nye et al., 2021). Since scratchpad only modifies the model's input and output (not the architecture), it is unclear and unexplored how architectural choices such as positional encoding affect the length generalization in the presence of scratchpad. To answer this question, we train all PEs *with* and *without* scratchpad on the mathematical and reasoning group of tasks, and compare their performance.

Moreover, the decision of how to represent the intermediate computations in the scratchpad, i.e. the scratchpad format, is an important design choice that has a non-trivial impact on the model's performance (Bueno et al., 2022).

To account for those, we consider five components in each step of scratchpad: `<input>`, `<computation>`, `<output>`, `<variable_update>`, and `<remaining_input>` (Figure 5). In our experiments, we create different variations of scratchpad format by enabling or disabling each component, which allows us to systematically study their impact.[1] Figure 6 summarizes our results. Similar to the remarks made by (Nye et al., 2021; Anil et al., 2022) we observe that across all PEs and regardless of the format, scratchpad is beneficial solely for the addition task. Additionally, our findings indicate that having a positional encoding with robust length generalization is crucial since scratchpad/CoT alone may not enhance the generalization.

```
Input (x)
Compute 5 3 7 2 6 + 1 9 1 7 =

Output (s; y)
<scratch>

 I    For digits 6 and 7,

 C    We have ( 6 + 7 + carry )
      % 10 = 13 % 10 = 13 % 10

 O    Which is equal to 3 .

 V    We update carry to 13 //
      10 = 1.

 R    So, the remaining input is
      5 3 7 2 + 1 9 1

...

</scratch>
The answer is 5 5 6 4 3.
```

Figure 5: Example of an addition task depicted with its first scratchpad step. Each step consists of five components: ▉ Step Input $\mathcal{I}$, ▉ Step Computation $\mathcal{C}$, ▉ Step Output $\mathcal{O}$, ▉ Intermediate Variable Updates $\mathcal{V}$, and ▉ Remaining Input $\mathcal{R}$.

### 6.1  Which part of the sequence is attended to?

The scratchpad format that is often used (Nye et al., 2021), similar to Figure 5, contains redundant information. One such example is the repetition of the remaining portion of an input ($\mathcal{R}$) in each step of the scratchpad. But, the attention can attend to this information directly from the main input. So, it remains unclear which specific part of the scratchpad different PEs rely on to solve the task.

To address this question, we take the models trained with full Format on addition, the case in which scratchpad is helpful across all PEs, and examine their attentions. Specifically, for tokens in

---

[1]Since using scratchpad creates very long sequences, we follow Nye et al. (2021) and set the length threshold $L = 8$ for tasks that use it to avoid out-of-memory errors.

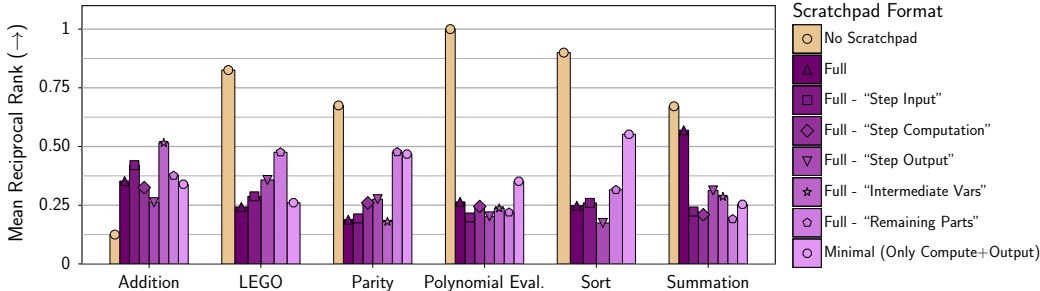

Figure 6: Mean ranks of scratchpad format aggregated across all models per each dataset. The effectiveness of scratchpad is task dependent.

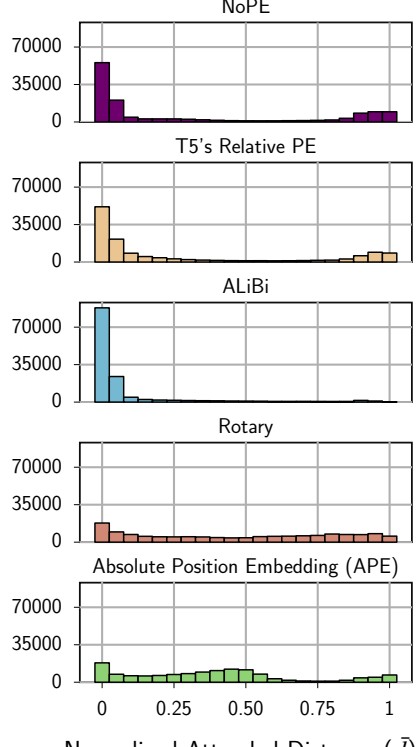

Figure 7: Distribution of the normalized distance between the query and the key of the self-attention (addition task + full scracthpad), averaged across all layers and heads.

the output sequence, we calculate the *distance* $d$ between the current query $q_t$ and the attended key $k_n$ as $(t - n + 1)$ and subsequently normalize it based on the length of the sequence at the present step. The normalized value is denoted as $\bar{d}$.

Figure 7 depicts the distribution of $\bar{d}$. Values of $\bar{d}$ close to 0 indicate attention to tokens near the current position (e.g. current scratchpad step), while values close to 1 signify attention to distant tokens (e.g. the input). NoPE and T5's Relative PE resemble each other and exhibit a bimodal distribution, reflecting both short-range and long-range attention. Conversely, ALiBi (due to its recency bias) strongly favors short-range attention. Rotary, on the other hand, produces a distribution resembling APE, which is more uniformly distributed. Notably, NoPE and T5's RPE are the top-performing PEs in this setup, which suggest the bimodal distribution to be more optimal.

## 7 Discussion

Practitioners have to make important choices about the nuances of the Transformer architecture like positional encoding before undertaking the costly pretraining process. In the I.I.D evaluation of PEs, we demonstrate similar performance across different PEs, in line with observations of Haviv et al. (2022) and Scao et al. (2022b), which makes the choice of optimal positional encoding challenging.

In our paper, we utilize length generalization in downstream tasks as a mean to assess the expressivity of positional encodings. Our setup, in contrast to the I.I.D. evaluation, reveals a clear distinction among approaches of encoding positions. We find that NoPE outperforms explicit PEs, and within explicit PEs, commonly used methods lag behind T5's Relative PE. In fact, the recent release of LLMs (Touvron et al., 2023; Chowdhery et al., 2022) suggests a shift towards adopting Rotary as a replacement for APE in the Transformer architecture. However, our result in Section 4 clearly demonstrates that Rotary marginally outperforms APE at length generalization. Furthermore, it exhibits similar behavior to APE, as shown in Section 6.1, indicating potential susceptibility to the same limitations.

The disadvantages of explicit PEs over NoPE in length extrapolation contribute to the growing evidence that positional encodings pose challenges for Transformers (Sinha et al., 2022; Luo et al., 2021). Our empirical results and theoretical analysis suggest that removing positional encoding holds promise as a modification to the widely used decoder-only Transformer architecture.

**Scaling up to 1B models**   In order to study the behavior of position embeddings at scale, we trained three 1B variants post-submission – ALiBi, Rotary and NoPE – with context length of 1024 tokens on a subset of StarCoder training set (Li et al., 2023). For more details, refer to Appendix F. Our results on language modeling show that at I.I.D all variants have similar perplexity, but at length generalization, Rotary fails to generalize as its perplexity explodes. NoPE and ALiBi generalize similarly to larger context sizes up to almost twice their training context size, and for larger contexts ALiBi is relatively more stable than NoPE (see the discussion on *perplexity vs. downstream performance*). Preliminary exploration of fine-tuning the pretrained models, on datasets in Section 3, yielded to identical performance among PE variants as the training context size of the 1.3B models is much larger than instance lengths in our datasets. A comprehensive downstream evaluation of these models remains a subject for future research.

**Perplexity vs. downstream Performance**   Due to human cognitive constraints (Gibson, 1998; Kiyono et al., 2021), language modeling data might encompasses short-range dependencies. The combination of this naturally occurring structure (which can be abundant in internet-based corpora) with the Recency Bias inherent in positional encodings like ALiBi could portray an unrealistic representation of models' length generalization performance. In fact, Chi et al. (2023) recently demonstrated that ALiBi's length generalization performance could be replicated using a window attention mask, where tokens beyond a window size $w$ are masked out. Interestingly, we also observe that T5's Relative PE, which can be regarded as trainable version of ALiBi, learns to attend both large and short range dependencies (Figure F.3). This is in line with Tay et al. (2022) observation and underscores the importance of evaluation setups on downstream tasks as compared to solely relying on perplexity.

# 8   Related Work

**Length Generalization Failure In Transformers**   The length generalization problem has been a topic of interest in the study of neural sequence models for a long time (Graves et al., 2016; Kaiser and Sutskever, 2016; Lake and Baroni, 2018; Hupkes et al., 2020; Yehudai et al., 2021). Transformers, being state-of-the-art sequence models, have been no exception. A group of studies showed the generalization failure of conventional Transformers with APE on specific datasets such as PCFG (Hupkes et al., 2020), LEGO (Zhang et al., 2023), or CLUTRR (Sinha et al., 2019; Gontier et al., 2020). The length generalization problem has been reported even in pretrained Transformers such as T5 (Furrer et al., 2020) and LaMDA (Anil et al., 2022). Csordás et al. (2021) and Ontanon et al. (2022) study the effect of positional encoding on length generalization but mainly focus on showing relative PE outperforms APEs. Press et al. (2022), on the other hand, propose a new encoding method, ALiBi, and demonstrate that it outperforms popular PEs on extrapolation but only in the context of human language modeling. Most relevant is Deletang et al. (2023)'s recent study on length generalization in various neural sequence models (including RNNs and Stacked-RNNs) for tasks from Chomsky hierarchy. However, they do not focus on the difference among positional encoding or on autoregressive models. Unlike these studies, our work extensively compares length generalization in popular PEs for a wide range of tasks, specifically focusing on autoregressive models, which represent many contemporary LLMs.

**Positional Encoding**   A core component of Transformers is the positional encoding mechanism, which helps the model represent the order of the input sequence. Self-attention mechanism in the encoder of Transformers is order-invariant and requires PE to avoid becoming a bag-of-word model. Many methods have been proposed for this purpose. Originally, Vaswani et al. (2017) introduced absolute positional encoding sinusoidal functions (a learned variant popularized by Devlin et al. (2019)). Relative approach for encoding positional information was further introduced by Shaw et al. (2018), which gave rise to a number of pre-trained LM with relative encodings such as TransformerXL (Dai et al., 2019) and T5 (Raffel et al., 2020) that perform well in length generalization. More recently, Su et al. (2021) takes the concept of sinusoidal functions and suggests a new way of encoding positional information by rotating the hidden representations before applying self-attention. This method, referred to as *Rotary*, has become a popular choice in the recent LLMs. Press et al. (2022) simplify the T5's Relative encoding and introduced a more efficient variant called ALiBi, while keeping the same or improving extrapolation performance. Decoder-only Transformers, due to their causal attention mask, are not order-agnostic and can operate without explicit positional information.

This was observed early on by Shen et al. (2018) and later explained by Tsai et al. (2019). The observation that Transformers without positional encoding can perform on par with explicit PE has been made in various domains such as machine translation (Yang et al., 2019), language modelling (Irie et al., 2019; Haviv et al., 2022), and even other domains like vision or speech (Likhomanenko et al., 2021). In our work, not only we demonstrate that Transformers can operate without explicit position information, but also we present an important setup where they outperform explicit PEs. Furthermore, we theoretically show how they are capable of learning both absolute and relative encodings.

## 9  Conclusion

We studied the robustness of different positional encodings, in decoder-only Transformers, at length generalization on various downstream mathematical and reasoning tasks. Our extensive empirical study shows the effectiveness of NoPE, and further demonstrates that widely used explicit PEs are not suited for length generalization. We also prove that NoPE can implicitly learn both absolute and relative positions, but uses the latter in practice. Finally, we find the effectiveness of scratchpad is task-dependent, and is not a reliable solution for length generalization.

## Limitations

Our work primarily focuses on positional encodings as a design choice in the Transformers decoder architecture. We could not study how large-scale pretraining affects different PEs because there are no publicly available large language models trained with various PEs under similar conditions. We leave this for future work due to our limited compute budget.

## Acknowledgements

We are grateful to our anonymous reviewers, Nicolas Chapados, and Omer Levy for their invaluable suggestions and discussions The Mila-IBM grant program provided the funding for this project. SR acknowledges the support provided by the NSERC Discovery Grant program and the Facebook CIFAR AI Chair program. This research was enabled in part by compute resources provided by Mila, the Digital Research Alliance of Canada and ServiceNow.

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

## A  Number of instances decreases rapidly as sequence length grows

The recent trend of SFT-RLHF pipeline (Ouyang et al., 2022) relies on finetuning LLMs on the instruction following tasks. However, the training data of these datasets is often skewed towards shorter sequences. Figure A.1 shows the distribution of instruction lengths in two instruction finetuning datasets: FLAN (CoT subset) (Longpre et al., 2023) and Super Natural Instructions (Wang et al., 2022). The median length of instructions in these datasets is quite short compared to the maximum length that exists. Such distribution shape highlights the importance of length generalization in these tasks. In fact, the models are supposed to learn from short instructions and generalize to ones during inference that might be much longer.

## B  Background

### B.1  Preliminaries

In this section, we lay the groundwork and introduce the notation we use throughout the paper. We will refer to this in Appendices C.1 and C.2.

Let $f_\theta$ be a decoder-only Transformer model, where $\theta$ denotes the full set of model parameters. $f_\theta$ processes the input sequence $x = [x_0, x_1, \ldots, x_T]$ and maps it to the output sequence $y = [y_0, y_1, \ldots, y_T]$ by applying a sequence of Transformer layers. Note that being decoder-only means the attention mechanism in each layer is causal, i.e. the attention weights are computed based on the previous positions only.

The layer $\text{TLayer}^{(l)}(\boldsymbol{H}^{(l-1)}; \theta_l)$, consisting of self-attention heads and a feed-forward sub-layer, reads the previous hidden state $\boldsymbol{H}^{(l-1)}$ and produces the hidden state at layer $l$: $\boldsymbol{H}^l$, where $l$ is the layer index, and $\theta_l$ is the set of parameters of the $l$-th layer. Each hidden state $\boldsymbol{H}^{(l)} \in \mathbb{R}^{d \times (T+1)}$ is matrix where column $t$, denoted as $\boldsymbol{h}_t^{(l)}$, is the hidden state at position $t$.

A layer $l$ is parameterized by a set of parameters $\theta_l = \{(\boldsymbol{W}_Q^m, \boldsymbol{W}_K^m, \boldsymbol{W}_V^m, \boldsymbol{W}_O^m)_m, \boldsymbol{W}_1, \boldsymbol{W}_2\}$, where $\boldsymbol{W}_Q^m, \boldsymbol{W}_K^m, \boldsymbol{W}_V^m \in \mathbb{R}^{h \times d}$ and $\boldsymbol{W}_O^m \in \mathbb{R}^{d \times h}$ are the query, key, value, and output matrices of the $m$-th head, respectively. $\boldsymbol{W}_1, \boldsymbol{W}_2 \in \mathbb{R}^{d \times k.d}$ are the weight matrices of the feed-forward sub-layer. $d$ denotes the model's hidden state size, $h$ is the attention dimension (where $h = \frac{d}{\# \text{heads}}$), and $k$ is a multiplier of the hidden state size in the feed-forward sub-layer (it is usually set to 4 in common implementations of the Transformer). Note that we drop the layer index $l$ and the attention head index $m$ where it is clear from the context.

The Transformer layer $\text{TLayer}^{(l)}$ processes each column of $\boldsymbol{H}^{(l-1)}$ independently and in parallel to produce the output. The computation of the $t$-th column of $\boldsymbol{H}^{(l)}$ is as follows:

$$\boldsymbol{h}_t^{(l)} = \text{FF}(\lambda(\boldsymbol{a}_t + \boldsymbol{h}_t^{(l-1)})) + \boldsymbol{a}_t + \boldsymbol{h}_t^{(l-1)} \tag{3}$$

where FF is the feed-forward sub-layer, $\lambda$ is layer normalization, and $\boldsymbol{a}_t \in \mathbb{R}^d$ is the output of the multi-head self-attention sub-layer at position $t$. Specifically, $\boldsymbol{a}_t$ is computed as:

$$\boldsymbol{a}_t = \sum_m \text{Attn}^{(m)}(\boldsymbol{h}_t^{(l-1)}, \boldsymbol{H}^{(l-1)}) \tag{4}$$

where $\text{Attn}^{(m)}$ is the $m$-th attention head. Let $\boldsymbol{o}_t \in \mathbb{R}^d$ denote the output of an attention head at position $t$. Then, $\boldsymbol{o}_t$ is computed as:

$$\boldsymbol{o}_t = \boldsymbol{W}_O \left( \sum_{i \le t} \hat{\boldsymbol{\alpha}}_i \boldsymbol{v}_i \right) \tag{5}$$

where $\hat{\boldsymbol{\alpha}} = \text{softmax}(\boldsymbol{\alpha}) \in \mathbb{R}^{(t+1)}$, and $\boldsymbol{\alpha}$ is the attention weight vector such that:

$$\boldsymbol{\alpha} = [\langle \boldsymbol{q}_t, \boldsymbol{k}_0 \rangle, \langle \boldsymbol{q}_t, \boldsymbol{k}_1 \rangle, \dots, \langle \boldsymbol{q}_t, \boldsymbol{k}_t \rangle]^\mathsf{T} \tag{6}$$

where $\boldsymbol{q}_t = \boldsymbol{W}_Q \boldsymbol{h}_t^{(l-1)} \in \mathbb{R}^h$, $\boldsymbol{k}_i = \boldsymbol{W}_K \boldsymbol{h}_i^{(l-1)} \in \mathbb{R}^h$, and $\boldsymbol{v}_i = \boldsymbol{W}_V \boldsymbol{h}_i^{(l-1)} \in \mathbb{R}^h$. $\langle \cdot, \cdot \rangle$ denotes the dot product operation.

The feed-forward sub-layer $\text{FF}(\cdot) \in \mathbb{R}^d$ is a two-layer MLP:

$$\text{FF}(\boldsymbol{x}) = \boldsymbol{W}_2 \sigma(\boldsymbol{W}_1^\mathsf{T} \boldsymbol{x}) \tag{7}$$

where $\sigma$ is a non-linear activation function (usually ReLU or GeLU (Hendrycks and Gimpel, 2020)). Additionally, $\lambda(\cdot) \in \mathbb{R}^d$ is layer normalization (Ba et al., 2016). Note that we take the *additive* (Elhage et al., 2021) view of attention heads in Equation (4) instead of *concatenate and multiple* view (Vaswani et al., 2017) as it is easier to understand and analyze. But, they are mathematically equivalent (Elhage et al., 2021).

The hidden state is initialized with a learned embedding of the input sequence $\boldsymbol{H}^{(0)} = \boldsymbol{W}_E \boldsymbol{X}$, where $\boldsymbol{W}_E \in \mathbb{R}^{d \times V}$ is the embedding matrix and $\boldsymbol{X} \in \mathbb{R}^{V \times (T+1)}$ is the one-hot encoded input sequence. $V$ is the vocabulary size.

## B.2 Positional Encoding

Almost all positional encoding methods can be explained and formulated as how they implement the dot product operation in Equation (6). So, in this section, we explain how the dot product $\langle \boldsymbol{q}_t, \boldsymbol{k}_i \rangle$ is implemented in different positional encoding schemes.

**Absolute Positional Encoding (APE)** The process of Absolute Positional Encoding (APE) involves assigning a position vector $\boldsymbol{p}_i$ to each absolute position $i$ and combining them with word embeddings before inputting them into the model. So, APE first modifies how the hidden state is initialized:

$$\boldsymbol{H}^{(0)} = \boldsymbol{W}_E \boldsymbol{X} + \boldsymbol{W}_P \boldsymbol{P} \tag{8}$$

where $\boldsymbol{W}_P \in \mathbb{R}^{d \times T}$ is the positional embedding matrix and $\boldsymbol{P} \in \mathbb{R}^{V_p \times (T+1)}$ is the one-hot encoded absolute position sequence. $V_p$ is the maximum absolute position. Therefore, the hidden state at column $j$ is:

$$\boldsymbol{h}_j^{(0)} = \boldsymbol{e}_j + \boldsymbol{p}_j \tag{9}$$

where $\boldsymbol{e}_j \in \mathbb{R}^d$ is the word embedding of token $x_j$ and $\boldsymbol{p}_j \in \mathbb{R}^d$ is the positional embedding for position $j$. Then, the dot product for the first layer in Equation (6) is computed as:

$$\begin{aligned}
\langle \boldsymbol{q}_t, \boldsymbol{k}_i \rangle &= \langle \boldsymbol{W}_Q \boldsymbol{h}_t^{(0)}, \boldsymbol{W}_K \boldsymbol{h}_i^{(0)} \rangle \\
&= \langle \boldsymbol{W}_Q(\boldsymbol{e}_t + \boldsymbol{p}_t), \boldsymbol{W}_K(\boldsymbol{e}_i + \boldsymbol{p}_i) \rangle \\
&= (\boldsymbol{W}_Q(\boldsymbol{e}_t + \boldsymbol{p}_t))^\mathsf{T} (\boldsymbol{W}_K(\boldsymbol{e}_i + \boldsymbol{p}_i)) \\
&= \boldsymbol{e}_t^\mathsf{T} \boldsymbol{W}_Q^\mathsf{T} \boldsymbol{W}_K \boldsymbol{e}_i + \boldsymbol{e}_t^\mathsf{T} \boldsymbol{W}_Q^\mathsf{T} \boldsymbol{W}_K \boldsymbol{p}_i \\
&\quad + \boldsymbol{p}_t^\mathsf{T} \boldsymbol{W}_Q^\mathsf{T} \boldsymbol{W}_K \boldsymbol{e}_i + \boldsymbol{p}_t^\mathsf{T} \boldsymbol{W}_Q^\mathsf{T} \boldsymbol{W}_K \boldsymbol{p}_i
\end{aligned} \tag{10}$$

In the learned variant of APE, $\boldsymbol{p}_j \in \mathbb{R}^d$ is learned during training. In the sinusoidal variant, $\boldsymbol{p}_j$ is calculated using a non-parametric function. Specifically, $\boldsymbol{p}_j$ is computed as:

$$\boldsymbol{p}_j = \left[\sin(\omega_1.j), \cos(\omega_1.j), \sin(\omega_2.j), \cos(\omega_2.j), \ldots, \sin(\omega_{d/2}.j), \cos(\omega_{d/2}.j)\right]^\mathsf{T} \quad (11)$$

where $\omega_i = \frac{1}{10000^{2i/d}}$.

**T5's Relative PE** The Relative bias in T5 is a type of relative positional encoding that initially calculates the relative distance $(t - i)$ between tokens at positions $t$ and $i$. This distance is then transformed into a scalar bias value $b$ and is incorporated into the dot product between the query and key. $b$ is learned during training. Thus, the dot product in every layer can be written as:

$$\langle \boldsymbol{q}_t, \boldsymbol{k}_i \rangle = \boldsymbol{q}_t^\mathsf{T} \boldsymbol{k}_i + b_{\mathrm{bucket}(n-m)} \quad (12)$$

where

$$\mathrm{bucket}(n) = \begin{cases} n & \text{if } n < \frac{\mathcal{B}}{2} \\ \frac{\mathcal{B}}{2} + \left\lfloor \frac{\log\left(\frac{n}{\mathcal{B}/2}\right)}{\log\left(\frac{\mathcal{D}}{\mathcal{B}/2}\right)} \times \frac{\mathcal{B}}{2} \right\rfloor & \text{if } \frac{\mathcal{B}}{2} \leq n < \mathcal{D} \\ \mathcal{B} - 1 & \text{if } n \geq \mathcal{D} \end{cases}$$

This function maps the relative distance $d$ to a bucket index, which will be used to look up the weight corresponding to that bucket. $\mathcal{B}$ is the number of buckets, $\mathcal{D}$ is the maximum distance. It assigns half of the buckets to distances smaller than $\frac{\mathcal{D}}{2}$ with linear spacing and the other half to distances larger than $\frac{\mathcal{D}}{2}$ with logarithmic spacing. The weight for distances larger than $\mathcal{D}$ is the same. This is to facilitate generalization to unseen distances. In the original implementation of T5, $\mathcal{B} = 32$ and $\mathcal{D} = 128$. Following shows an example of the bucket function with $\mathcal{B} = 5$ and $\mathcal{D} = 6$:

$$\mathrm{bucket}\left(\begin{bmatrix} 0 & 0 & 0 & 0 & 0 & 0 & 0 & 0 & 0 & 0 \\ 1 & 0 & 0 & 0 & 0 & 0 & 0 & 0 & 0 & 0 \\ 2 & 1 & 0 & 0 & 0 & 0 & 0 & 0 & 0 & 0 \\ 3 & 2 & 1 & 0 & 0 & 0 & 0 & 0 & 0 & 0 \\ 4 & 3 & 2 & 1 & 0 & 0 & 0 & 0 & 0 & 0 \\ 5 & 4 & 3 & 2 & 1 & 0 & 0 & 0 & 0 & 0 \\ 6 & 5 & 4 & 3 & 2 & 1 & 0 & 0 & 0 & 0 \\ 7 & 6 & 5 & 4 & 3 & 2 & 1 & 0 & 0 & 0 \\ 8 & 7 & 6 & 5 & 4 & 3 & 2 & 1 & 0 & 0 \\ 9 & 8 & 7 & 6 & 5 & 4 & 3 & 2 & 1 & 0 \end{bmatrix}\right) = \begin{bmatrix} 0 & 0 & 0 & 0 & 0 & 0 & 0 & 0 & 0 & 0 \\ 1 & 0 & 0 & 0 & 0 & 0 & 0 & 0 & 0 & 0 \\ 2 & 1 & 0 & 0 & 0 & 0 & 0 & 0 & 0 & 0 \\ 3 & 2 & 1 & 0 & 0 & 0 & 0 & 0 & 0 & 0 \\ 3 & 3 & 2 & 1 & 0 & 0 & 0 & 0 & 0 & 0 \\ 4 & 3 & 3 & 2 & 1 & 0 & 0 & 0 & 0 & 0 \\ 4 & 4 & 3 & 3 & 2 & 1 & 0 & 0 & 0 & 0 \\ 4 & 4 & 4 & 3 & 3 & 2 & 1 & 0 & 0 & 0 \\ 4 & 4 & 4 & 4 & 3 & 3 & 2 & 1 & 0 & 0 \\ 4 & 4 & 4 & 4 & 4 & 3 & 3 & 2 & 1 & 0 \end{bmatrix}$$

**ALiBi** Similar to T5's Relative PE, ALiBi subtracts a scalar bias from the attention score. As the distance between the query and key tokens increases, the bias grows linearly. Specifically, the dot product in every layer can be written as:

$$\langle \boldsymbol{q}_t, \boldsymbol{k}_i \rangle = \boldsymbol{q}_t^\mathsf{T} \boldsymbol{k}_i - (t - i).C^{(m+1)} \quad (13)$$

where $m$ is head index and $C$ is a constant defined as:

$$C = 2^{-2^{-\log_2(\#\,\text{heads}+3)}}$$

For example, if the number of heads is 8, then we have $\frac{1}{2}, \frac{1}{2^2}, \ldots, \frac{1}{2^8}$ (Press et al., 2022).

**Rotary** The Rotary is a relative PE that applies a rotation to the query and key representations based on their absolute positions before dot product attention. Due to this rotation, the attention dot product relies solely on the relative distance between tokens.

First, we formulate Rotary for model dimension $d = 2$. Rotary positional encoding defines the dot product as:

$$
\begin{aligned}
\langle \boldsymbol{q}_t, \boldsymbol{k}_i \rangle &= \langle \mathrm{Rot}(\boldsymbol{q}_t, t), \mathrm{Rot}(\boldsymbol{k}_i, i) \rangle \\
&= \langle \boldsymbol{R}^{t\theta} \boldsymbol{q}_t, \boldsymbol{R}^{i\theta} \boldsymbol{k}_i \rangle \\
&= (\boldsymbol{R}^{t\theta} \boldsymbol{q}_t)^\intercal (\boldsymbol{R}^{i\theta} \boldsymbol{k}_i) \\
&= \boldsymbol{q}_t^\intercal (\boldsymbol{R}^{t\theta})^\intercal \boldsymbol{R}^{i\theta} \boldsymbol{k}_i \\
&= \boldsymbol{q}_t^\intercal \boldsymbol{R}^{(i-t)\theta} \boldsymbol{k}_i
\end{aligned}
$$

$$(14)$$

where $\boldsymbol{R}^{t\theta}$ is a rotation matrix that rotates $\boldsymbol{x}$ by $t\theta$ radians:

$$
\boldsymbol{R}^{n\theta} = \begin{bmatrix} \cos(n\theta) & -\sin(n\theta) \\ \sin(n\theta) & \cos(n\theta) \end{bmatrix} \tag{15}
$$

for $d > 2$, Rotary applies the same approach on every two consecutive dimensions of $\boldsymbol{q}_t$ and $\boldsymbol{k}_i$, but with different $\theta$ angles. Refer to Su et al. (2021) for the exact formulation.

**NoPE** NoPE does not explicitly encode positional encodings. So, the dot product in every layer can be written as:

$$
\langle \boldsymbol{q}_t, \boldsymbol{k}_i \rangle = \boldsymbol{q}_t^\intercal \boldsymbol{k}_i \tag{16}
$$

## C  Proofs

In this section, we provide proof of why NoPE can implicitly learn both absolute and relative positions. We refer the readers to Appendix B.1 for the notation and definitions used in this section.

### C.1  Absolute Positional Encoding in NoPE

This section discusses how NoPE can recover absolute positions in the hidden state. Our proof is inspired by Weiss et al. (2021); Lindner et al. (2023) and relies on the causal attention mask in the decoder-only Transformer and the $\mathrm{softmax}$ function to recover absolute positions.

> **Theorem 1** (**Absolute Encoding**). *Let $\boldsymbol{x} = [\texttt{<bos>}, x_1, \ldots, x_T]$ be an input sequence of length $T + 1$ to the model. Then, the first layer of $f_\theta$ can recover absolute positions $[1, \ldots, T + 1]$ in the hidden state $\boldsymbol{H}^{(1)}$. That is, there exist $\boldsymbol{W}_Q$, $\boldsymbol{W}_K$, $\boldsymbol{W}_V$, $\boldsymbol{W}_O$, $\boldsymbol{W}_1$, and $\boldsymbol{W}_2$ such that the self-attention and feedforward operations in the first layer compute absolute positions and write it to the next hidden state.*

*Proof.*

Our proof only specifies the weights of a single attention head in the first layer (and additionally the parameterization of feedforward sub-layer). In this parameterization, we only require the first three dimensions of the hidden states. The rest of the heads, as long as they do not override the first three dimensions, can be arbitrary. This does not impose any challenges as Transformers used in practice usually have a very large model dimension $d$. In the rest, we provide the construction of the weights and then verify that they can recover absolute positions.

First, we construct the word embedding matrix $\boldsymbol{W}_E \in \mathbb{R}^{d \times V}$, where each column is the embedding of a token in the vocabulary. We construct $\boldsymbol{W}_E$ such that it always sets the first dimension of every embedding vector to be 1. Additionally, it sets the second dimension to 1 if and only if the token is <bos>. Otherwise, it sets it to zero. The third dimension of all embedding vectors is set to zero. Other dimensions can take any arbitrary values. Without loss of generality, assume <bos> is the first token in the vocabulary, i.e. The first column. Then, we have:

$$
\boldsymbol{W}_E = \begin{bmatrix}
1 & 1 & 1 & \cdots & 1 \\
1 & 0 & 0 & \cdots & 0 \\
0 & 0 & 0 & \cdots & 0 \\
e_{4,1} & e_{4,2} & e_{4,3} & \cdots & e_{4,V} \\
\vdots & \vdots & \vdots & \ddots & \vdots \\
e_{d,1} & e_{d,2} & e_{d,2} & \cdots & e_{d,V}
\end{bmatrix}_{d \times V} \tag{17}
$$

where $e_{d,i} \in \mathbb{R}$.

Secondly, for head dimensions $h \geq 1$, we construct the weights $\boldsymbol{W}_Q, \boldsymbol{W}_K, \boldsymbol{W}_V, \boldsymbol{W}_O$ of the first attention head in the first layer. Specifically,

$$
\boldsymbol{W}_K = \begin{bmatrix} 1 & 0 & \cdots & 0 \\ 1 & 0 & \cdots & 0 \\ \vdots & \vdots & \ddots & \vdots \\ 1 & 0 & \cdots & 0 \end{bmatrix}_{h \times d} \qquad \boldsymbol{W}_V = \begin{bmatrix} 0 & 1 & 0 & \cdots & 0 \\ 0 & 0 & 0 & \cdots & 0 \\ \vdots & \vdots & \vdots & \ddots & \vdots \\ 0 & 0 & 0 & \cdots & 0 \end{bmatrix}_{h \times d} \tag{18}
$$

$\boldsymbol{W}_K$ reads from the first dimension of the hidden state, which is initialized with 1 using the embedding matrix. Since all word embeddings have one in their first dimension, this parameterization will result all key vectors to be the same. Moreover, $\boldsymbol{W}_V$ reads from the second dimension of the hidden state, which is initialized with 1 if the token is <bos>. So, the value vector will have 1 in its first dimension only if the corresponding token is <bos>.

$\boldsymbol{W}_Q$ can be any arbitrary matrix. $\boldsymbol{W}_O$ will write the result of the attention to the third dimension of the hidden state and can be constructed as:

$$
\boldsymbol{W}_O = \begin{bmatrix} 0 & 0 & 0 & 0 & \cdots & 0 \\ 0 & 0 & 0 & 0 & \cdots & 0 \\ 1 & 0 & 0 & 0 & \cdots & 0 \\ 0 & 0 & 0 & 0 & \cdots & 0 \\ \vdots & \vdots & \vdots & \vdots & \ddots & \vdots \\ 0 & 0 & 0 & 0 & \cdots & 0 \end{bmatrix}_{d \times h} \tag{19}
$$

Now, we verify that for any input sequence $\boldsymbol{x} = [\texttt{<bos>}, x_1, \ldots, x_T]$, the first layer can recover absolute positions $[1, \ldots, T+1]$ in the hidden state $\boldsymbol{H}^{(1)}$. We verify this for column $t$ of $\boldsymbol{H}^{(1)}$. That is, we show that absolute position information is available in the third dimension of $\boldsymbol{h}_t^{(1)}$.

First, we use the word embedding matrix $\boldsymbol{W}_E$ to compute the embedding $\boldsymbol{H}^{(0)}$:

$$
\boldsymbol{H}^{(0)} = \boldsymbol{W}_E \boldsymbol{X} = \begin{bmatrix} 1 & 1 & 1 & \cdots & 1 \\ 1 & 0 & 0 & \cdots & 0 \\ 0 & 0 & 0 & \cdots & 0 \\ e_{4,1} & e_{4,2} & e_{4,3} & \cdots & e_{4,V} \\ \vdots & \vdots & \vdots & \ddots & \vdots \\ e_{d,1} & e_{d,2} & e_{d,2} & \cdots & e_{d,V} \end{bmatrix}_{d \times (T+1)} \tag{20}
$$

We now provide the attention computation at position $1 \leq t \leq T+1$. First, we use $\boldsymbol{W}_Q$ to compute the query vector $\boldsymbol{q}_t$ by applying $\boldsymbol{q}_t = \boldsymbol{W}_Q \boldsymbol{h}_t^{(0)}$:

$$
\boldsymbol{q}_t = [q_1, q_2, q_3, \ldots, q_h]^\mathsf{T} \tag{21}
$$

Recall that $\boldsymbol{W}_Q$ can be any arbitrary matrix. So, $q_j \in \mathbb{R}$ can take any arbitrary value. Next, we compute the key vectors by applying $\boldsymbol{k}_i = \boldsymbol{W}_K \boldsymbol{h}_i^{(0)}$:

$$
\boldsymbol{k}_1 = \begin{pmatrix} 1 \\ 1 \\ \vdots \\ 1 \end{pmatrix} \qquad \boldsymbol{k}_2 = \begin{pmatrix} 1 \\ 1 \\ \vdots \\ 1 \end{pmatrix} \qquad \cdots \qquad \boldsymbol{k}_t = \begin{pmatrix} 1 \\ 1 \\ \vdots \\ 1 \end{pmatrix} \tag{22}
$$

Note that all key vectors are the same and we only need to compute them up to position $t$ as the attention mask is causal, i.e query can only look at positions $\leq t$. Next, we compute the attention weight vectors $\boldsymbol{\alpha}$:

$$
\boldsymbol{\alpha} = [\langle \boldsymbol{q}_t, \boldsymbol{k}_1 \rangle, \langle \boldsymbol{q}_t, \boldsymbol{k}_2 \rangle, \ldots, \langle \boldsymbol{q}_t, \boldsymbol{k}_t \rangle]^\mathsf{T} \tag{23}
$$

$$
= [\alpha^*, \alpha^*, \ldots, \alpha^*]^\mathsf{T} \tag{24}
$$

where $\alpha^* = q_1 + q_2 + \ldots + q_h$. Next, we apply $\mathrm{softmax}$ to compute the attention probabilities. Since all $\boldsymbol{\alpha}^i$'s are the same, we have:

$$\hat{\boldsymbol{\alpha}} = \mathrm{softmax}(\boldsymbol{\alpha}) = \left[\frac{1}{t}, \frac{1}{t}, \ldots, \frac{1}{t}\right]^{\mathsf{T}} \tag{25}$$

Now, we compute the value vectors by applying $\boldsymbol{v}_i = \boldsymbol{W}_V \boldsymbol{h}_i^{(0)}$:

$$\boldsymbol{v}_1 = \begin{pmatrix} 1 \\ 0 \\ \vdots \\ 0 \end{pmatrix} \quad \boldsymbol{v}_2 = \begin{pmatrix} 0 \\ 0 \\ \vdots \\ 0 \end{pmatrix} \quad \ldots \quad \boldsymbol{v}_t = \begin{pmatrix} 0 \\ 0 \\ \vdots \\ 0 \end{pmatrix} \tag{26}$$

Finally, we compute the output of the attention head by applying $\boldsymbol{W}_O$:

$$\boldsymbol{o}_t = \boldsymbol{W}_O \left(\sum_{i \le t} \hat{\boldsymbol{\alpha}}_i \boldsymbol{v}_i\right) = \boldsymbol{W}_O \left(\frac{1}{t} \sum_{i \le t} \boldsymbol{v}_i\right) = \boldsymbol{W}_O \begin{pmatrix} 1/t \\ 0 \\ \vdots \\ 0 \end{pmatrix}_h = \begin{pmatrix} 0 \\ 0 \\ 1/t \\ 0 \\ \vdots \\ 0 \end{pmatrix}_d \tag{27}$$

Thus, the output of our constructed attention head recovers the absolute position information and writes it to the third dimension of output.

We used the decoder-only property of Transformer implicitly in Equation (23), which helped us to only attend to position $\le t$. So, the lengths of the attended sequence are always $t$. Moreover, the presence of `<bos>` token in the input sequence helped us to anchor the absolute position information. This is not a problem as in practice models are often prompted with some instructions which can act as `<bos>` token.

With this information available to the rest of the network, the feedforward sub-layer, with sufficient hidden width, can recover the absolute positions $[1, 2, \ldots, T+1]$ from the third dimension of attention output. This is because the feedforward sub-layer is MLP with ReLU activation. So, it can learn any arbitrary function (Park et al., 2020). Note that the layer-norm operation can be bypassed as explained by Akyurek et al. (2023). □

## C.2 Relative Positional Encoding in NoPE

In this section, we show if the hidden state contains absolute positional information as explained in the previous section, then the attention mechanism in all subsequent layers can implement a relative positional encoding. We refer the readers to Appendices B.1 and C.1 for the notation and definitions used in this section.

> **Theorem 2 (Relative Encoding).** *Suppose that the hidden state $\boldsymbol{H}^{(1)}$ contains absolute positional information, as stated in Theorem 1, and assume that it is not overwritten by any subsequent layers. Then, the self-attention in all subsequent layers can implement a relative positional encoding: there exists a parameterization of $f_\theta$ such that, for $l \ge 2$, the attention dot product between query $\boldsymbol{q}_t$ and key $\boldsymbol{k}_i$ at positions $t$ and $i$ ($t \ge i$) can be expressed as:*
>
> $$\langle \boldsymbol{q}_t, \boldsymbol{k}_i \rangle = f_{\mathrm{cnt}}(\boldsymbol{q}_t, \boldsymbol{k}_i) + f_{\mathrm{rel}}(t - i) \tag{1}$$
>
> *where $f_{\mathrm{cnt}}$ is a function of their content, and $f_{\mathrm{rel}}$ is a function of their relative distance.*

*Proof.*

Our proof only specifies a few entries of weight matrices for attention heads in layers $l \ge 2$, which does not impose any challenges for Transformers used in practice as they usually have a very large model dimension $d$. Moreover, we require to have absolute positions in the third dimension of the hidden state as explained in Theorem 1. To show NoPE can implement relative encoding, we only need to prove that its attention dot product depends on the relative distance between tokens (See

Appendix B.1 for an overview of relative encoding methods). In the rest, we provide the construction of the weights and then verify that they can implement relative position encoding.

For head dimension $h \geq 2$, we construct the weights $\boldsymbol{W}_Q, \boldsymbol{W}_K$ of the attention heads in the second layers and above. Specifically,

$$
\boldsymbol{W}_Q = \begin{bmatrix}
1 & 0 & 0 & 0 & \dots & 0 \\
0 & 0 & -1 & 0 & \dots & 0 \\
w_{3,1} & w_{3,2} & w_{3,3} & w_{3,4} & \dots & w_{3,d} \\
\vdots & \vdots & \vdots & \vdots & \ddots & \vdots \\
w_{h,1} & w_{h,2} & w_{h,3} & w_{h,4} & \dots & w_{h,d}
\end{bmatrix}_{h \times d}
\tag{28}
$$

$$
\boldsymbol{W}_V = \begin{bmatrix}
0 & 0 & 1 & 0 & \dots & 0 \\
1 & 0 & 0 & 0 & \dots & 0 \\
w'_{3,1} & w'_{3,2} & w'_{3,3} & w'_{3,4} & \dots & w'_{3,d} \\
\vdots & \vdots & \vdots & \vdots & \ddots & \vdots \\
w'_{h,1} & w'_{h,2} & w'_{h,3} & w'_{h,4} & \dots & w'_{h,d}
\end{bmatrix}_{h \times d}
\tag{29}
$$

where $w_{i,j}, w'_{i,j} \in \mathbb{R}$ can take any arbitrary value. Their corresponding $\boldsymbol{W}_V$ and $\boldsymbol{W}_O$ can take any arbitrary values as long as they do not override the first three dimensions of the residual stream.

Now we verify that for any input sequence $\boldsymbol{x} = [\texttt{<bos>}, x_1, \dots, x_T]$, the attention dot product between query $\boldsymbol{q}_t$ and key $\boldsymbol{k}_i$ at positions $t$ and $i$ ($t \geq i$) will depend the relative distance between tokens.

First, assume that absolute positions are computed in the hidden state $\boldsymbol{H}^{(l)}$ for $l \geq 1$, as stated in Theorem 1. Specifically,

$$
\boldsymbol{H}^{(l)} = \begin{bmatrix}
1 & 1 & 1 & 1 & \dots & 1 \\
1 & 0 & 0 & 0 & \dots & 0 \\
1 & 2 & 3 & 4 & \dots & T+1 \\
h_{4,1} & h_{4,2} & h_{4,3} & h_{4,4} & \dots & h_{4,T+1} \\
\vdots & \vdots & \vdots & \vdots & \ddots & \vdots \\
h_{d,1} & h_{d,2} & h_{d,3} & h_{d,4} & \dots & h_{d,T+1}
\end{bmatrix}_{d \times (T+1)}
\tag{30}
$$

where $h_{i,j} \in \mathbb{R}$ can be any arbitrary value as the first three dimensions of the hidden state are reserved for PE computation. The rest of the dimensions can take any arbitrary values as in regular computation of Transformers.

We now present the attention computations at position $1 \leq t \leq T+1$. We use $\boldsymbol{W}_Q$ to compute the query vector $\boldsymbol{q}_t$ by applying $\boldsymbol{q}_t = \boldsymbol{W}_Q \boldsymbol{h}_t^{(l)}$:

$$
\boldsymbol{q}_t = [1, -t, q_3, \dots, q_h]^\mathsf{T}
\tag{31}
$$

where $q_j \in \mathbb{R}$ can take any arbitrary value. Next, we compute the key vectors by applying $\boldsymbol{k}_i = \boldsymbol{W}_K \boldsymbol{h}_i^{(l)}$:

$$
\boldsymbol{k}_1 = \begin{pmatrix} 1 \\ 1 \\ k_{3,1} \\ \vdots \\ k_{h,1} \end{pmatrix} \quad
\boldsymbol{k}_2 = \begin{pmatrix} 2 \\ 1 \\ k_{3,2} \\ \vdots \\ k_{h,2} \end{pmatrix} \quad
\boldsymbol{k}_3 = \begin{pmatrix} 3 \\ 1 \\ k_{3,3} \\ \vdots \\ k_{h,3} \end{pmatrix} \quad \dots \quad
\boldsymbol{k}_t = \begin{pmatrix} t \\ 1 \\ k_{3,t} \\ \vdots \\ k_{h,t} \end{pmatrix}
\tag{32}
$$

where $k_{(\cdot,\cdot)} \in \mathbb{R}$ can have any arbitrary value. So, for $\boldsymbol{k}_i$ we have:

$$
\boldsymbol{k}_i = [i, 1, k_{3,i}, \dots, k_{h,i}]^\mathsf{T}
\tag{33}
$$

Next, we let us present the attention dot product between $\boldsymbol{q}_t$ and $\boldsymbol{k}_i$:

$$\langle \boldsymbol{q}_t, \boldsymbol{k}_i \rangle = 1.i + (-t).1 + q_3.k_{3,i} + \cdots + q_h.k_{h,i} \tag{34}$$

$$= i - t + \sum_{j=3}^{h} q_j.k_{j,i} \tag{35}$$

$$= \left( \sum_{j=3}^{h} q_j.k_{j,i} \right) - (t - i) \tag{36}$$

$$= f_{\text{cnt}}(\boldsymbol{q}_t, \boldsymbol{k}_i) + f_{\text{rel}}(t - i) \tag{37}$$

Thus, the dot product between $\boldsymbol{q}_t$ and $\boldsymbol{k}_i$ depends on the relative distance between tokens (assuming the rest of the terms do not cancel out which can be easily avoided by setting the respective weights in Equations (28) and (29)). Note that our proof uses the linear spacing between tokens, but the MLP the first layer can write any arbitrary function of absolute positions to the third dimension of the hidden state, which enables more complex relative encoding schemes. □

## D  Experimental Details

### D.1  Tasks

Here we provide the details and more examples of the tasks and datasets we used in our evaluation. For each task, we sample 100K examples for the training set and 10K for the test. Also, we use 15% of the train as the validation set.

**Addition**  The addition task (Nye et al., 2021) asks the model to compute the sum of two numbers. Each number is represented as a sequence of digits that are separated by space. So, the model has access to the exact digits.

> Input
>
> `Compute:  5 3 7 2 6 + 1 9 1 7 ?`
>
> Output
>
> `The answer is 5 5 6 4 3.`

we create each length bucket based on the number of digits in each number, e.g. 6-by-3, 6-by-4, etc. For the training set, we use buckets where one of the numbers has at most $L$ digits. For the test set, we use buckets where any of the numbers have at most $L$ digits. The model is evaluated on the correctness of its predicted result.

**Polynomial Evaluation**  The polynomial evaluation task (Nye et al., 2021) asks the model to evaluate a polynomial expression at a given value of $x$. The polynomial terms and digits are separated to make just the tokenizer does not glue symbols together.

> Input
>
> `Evaluate x = 3 in ( 3 x ** 0 + 1 x ** 1 + 1 x ** 2 ) % 10 ?`
>
> Output
>
> `The answer is 5.`

The length bucket is created based on the number of terms in the polynomial expression. We sample $x$ from $\mathcal{U}(-2, 2)$, the degree of each term from $\mathcal{U}(0, 3)$, and the coefficient of each term from $\mathcal{U}(-3, 3)$. We take the modulo of the result by 10 to make the task easier for the model and make sure we only measure the generalization of the length of the problem instance not the value of the polynomial. The model is evaluated on the correctness of its predicted result.

**Sorting**  The sorting task (Saxton et al., 2019) asks the model to sort a sequence of input numbers. We use this task in two variants: Single Token and Multi Digit. In the Single Token variant, we create an alphabet of 50 tokens from the model's vocabulary and fix some canonical ordering among them

through task. Each instance is a sequence of tokens from the alphabet in a random order, and the model is asked to sort them in the canonical order.

> Input
>
> `Sort the following numbers:  3 1 4 1 5 ?`
>
> Output
>
> `The answer is 1 1 3 4 5.`

In the Multi Digit variant, we simply present a sequence of multi digit/tokens numbers to the model, and ask it to sort them in ascending order. Each number is represented by its digits and they are separated by a space.

> Input
>
> `Sort the following numbers:  3 1, 4 1, 5 9, 1 2 6, 5 3 3 ?`
>
> Output
>
> `The answer is 3 1, 4 1, 5 9, 1 2 6, 5 3 3.`

In this case, we sample each number from $\mathcal{U}(0, 10000)$. In both cases, the length bucket is created based on the length of the input sequence. The model is evaluated on the correctness of its predicted result.

**Summation**   In this task (Saxton et al., 2019), we ask the model to compute the sum of a sequence of input numbers modulo 10 as we want to specifically measure how the model generalizes to longer sequences not the value of summation result:

> Input
>
> `Compute:  ( 1 + 2 + 3 + 4 + 7 ) % 10 ?`
>
> Output
>
> `The answer is 7.`

Each digit is randomly sampled from $\mathcal{U}(1, 9)$. The length bucket is created based on the length of the input sequence. The model is evaluated on the correctness of its predicted result.

**Parity**   In the parity task (Anil et al., 2022), we ask the model to compute the parity of a binary sequence.

> Input
>
> `Is the number of 1's even in [ 1 0 0 1 1] ?`
>
> Output
>
> `The answer is No.`

**LEGO**   In the LEGO task (Zhang et al., 2023), the model is provided with a simple computation graph (DAG), where each node represents a variable, and variables are connected by simple operations which created the edges in the computation graph. We refer to Zhang et al. (2023) for a detailed description.

> Input
>
> `If a = -1; b = -a; c = +b; d = +c.  Then what is c?`
>
> Output
>
> `The answer is +1.`

To sample each example, we first sample the list of variables based on the length of the example, and then we uniformly sample the value of each variable to make sure all variables are represented with both -1 and +1. Finally, given the value of variables, we deterministically compute the operation on each edge. For each example, we query all variables from the middle of the computation graph to the end. The model is evaluated on the correctness of its predicted result.

**Copy** The copy task is straightforward. The model has to repeat the input sequence in the output.

Input

```
Copy the following words:  <w1> <w2> <w3> <w4> <w5> .
```

Output

```
<w1> <w2> <w3> <w4> <w5>
```

We create multiple variants of this task to better understand the models' generalization behavior. In the first variant, the input tokens are the same, so the model has to basically count the number of input tokens. In the second variant, the model has to replace the input tokens with another token sampled from the vocabulary. In the third variant, we sample the input tokens from the model's vocabulary, and the model has to predict them in the same order. We also create 2x versions of variants 1 and 3 to make the tasks more challenging.

**Reverse** In this task the model, the model has to reverse the order of input tokens in its output.

Input

```
Reverse the following words:  <w1> <w2> <w3> <w4> <w5> .
```

Output

```
<w5> <w4> <w3> <w2> <w1> .
```

As in the copy task, we create multiple variants of this task. In the first variant, the model has to reverse the order of input tokens, where the tokens are randomly sampled from the model's vocabulary. In the second variant, the model has to reverse the order of input tokens, as in the first variant, but also it has to reverse it one more time, recreating the original input.

### D.2  Hyperparameters

Table 2 shows the hyperparameters we used in our experiments. We use the same hyperparameters for all models and positional encoding schemes. In our initial experiment, we tried a few more hyperparameters such as $\mathrm{lr} \in \{0.00001, 0.00003, 0.00005\}$ and $\mathrm{WeightDecay} \in \{0, 0.05, 0.1\}$, but we did not observe any significant difference in the results. So, we decided to use the same hyperparameters throughout our experiments.

### D.3  Compute

In our experiments, we used single-GPU training setup for the models. Specifically, we ran our experiments on a mix of NVIDIA V100 32G, NVIDIA RTX8000 48G, NVIDIA A100 40G, and NVIDIA A100 80G GPUs. Depending on the GPU type and the positional encoding, each of our training runs took 6 to 15 hours, per each seed, on average to complete. Considering all the datasets, and positional encoding schemes, in addition to the scratchpad experiments, and three seeds, we ran about 870 individual training runs for the results in this paper.

### D.4  Reproducibility

In this study, all experiments employed open-source libraries, specifically HuggingFace (Wolf et al., 2020) from which we utilized their implementation as a foundation for the training loop, optimizer, and the Transformer architecture. To ensure reproducibility, we will also release a singularity binary with all dependencies and libraries to enable running our experiments on any machine with NVIDIA GPUs and at any time in the future. Moreover, every reported number in this paper is linked to the source code package that deterministically (up to GPU stochasticity) reproduces the results, which we release publicly on GitHub at `https://github.com/McGill-NLP/length-generalization`.

## E  Full Results

### E.1  Detailed Model Accuracy

We report the detailed results of our experiments in Figures F.4 to F.6. We refer the readers to Appendix D.1 for the description of each task.

Table 2: Summary of hyperparamters used in the experiments.

| Parameter | Value |
|---|---|
| Optimizer | AdamW |
| Learning rate | 0.00003 |
| Weight Decay | 0.05 |
| Batch size | 64 |
| Learning Rate Scheduler | Polynomial |
| Warm Up | 6% of training steps |
| # Train Steps | 40K steps |
| Decoding Method | Greedy (No Sampling) |
| Dropout *(taken from HuggingFace)* | 0.1 |
| Model dimension *(taken from HuggingFace)* | 768 |
| # Layers *(taken from HuggingFace)* | 12 |
| # Attention Heads *(taken from HuggingFace)* | 12 |

### E.2  Detailed Head Distance

Figure F.7 shows the layer-wise distance of No PE's attention patterns with other positional encoding schemes measured across instances of the SCAN dataset. We refer the readers to Section 5.2 for the details and analysis of these results.

### E.3  Detailed Model Accuracy On Various Scratchpad Formats

Figure F.8 shows the generalization of various scratchpad formats for each model aggregated across all datasets. Figures F.9 to F.15 show the generalization of various scratchpad formats for each model on each dataset. We refer the readers to Section 6 for the details and analysis of these results.

## F  Pretraining at 1.3B Scale

In this section, we elucidate our preliminary assessment of length generalization on a pretrained LLM scaled to 1.3B. Ensuring a fair evaluation, we pretrain models across varied positional encodings, all on identical data with consistent parameters. Given the intrinsic significance of element positions to the semantics in code data, our choice leaned towards code-based pretraining.

### F.1  Model Architecture

Our approach remains consistent with the original codebase, merely increasing the model dimensions. Specifically, we adopt a decoder-only Transformer structure encompassing 24 layers. The configuration details are as follows: $d_{\text{model}} = 1024$, $d_{\text{kv}} = 128$, $d_{\text{ff}} = 16,384$, and 32 attention heads. Consequently, this amounts to a Transformer model with 1.3B parameters. We train the models with a context size of 1024 tokens. For the rest of training hyperparameters, we follow Allal et al. (2023). Specifially, we use a global batch size of 256 and AdamW optimizer with $\beta_1 = 0.9$, $\beta_2 = 0.95$, $\epsilon = 10^{-8}$, and weight decay 0.1. Also, the learning rate of $2 \times 10^{-4}$ with cosine decay and 2% warm-up is used.

### F.2  Dataset Selection

We utilize a subset of 30M documents from the StarCoder dataset (Li et al., 2023), constituting a blend of 40% Python, 25% Java, 25% JavaScript, 5% GitHub issues, and 5% GitHub commits. Processed through the StarCoder tokenizer with a vocab size of $49,152$, the dataset results in 30B tokens. We have trained the model for a single epoch. For comprehensive details on the remaining hyperparameters, we direct readers to Allal et al. (2023).

### F.3 Generalization Evaluation

Our evaluation aims to assess how the model's performance in predicting a given token varies with changes in context size, especially when the context size exceeds the training context size. Ideally, we expect the perplexity to generally improve as the context size increases since the model has access to more information. In this experiment, we select a sample of $1500$ validation documents, each representing a single source code file. For each document, we calculate the perplexity of the last tokens, considering various context sizes ranging from $512$ to $2560$ (Note the model itself is pretrained with a context size of $1024$) This approach allows us to evaluate the model's perplexity under both in-distribution and out-of-distribution conditions. To keep the computation and number of inferences tractable, we measure the perplexity on a maximum of last 200 tokens in the document and ensure exactly the same set of tokens are evaluated per each context size.

To obtain a comprehensive understanding of the model's performance, we also categorize documents into buckets based on their lengths and calculate the average perplexity within each bucket. This categorization enables us to examine the nature of dependencies within each bucket, recognizing that longer documents do not always imply longer dependencies.

We present the results in Figure F.2 and Tables 3 and 4. First, it is observed that all models benefit from a larger context up to the training size of $1024$. In the I.I.D. case (context length $\leq 1024$), no significant differences among the models are discernible across all length buckets. Conversely, in the O.O.D. case (context length $> 1024$), the model trained with Rotary fails to generalize as its perplexity explodes in all cases, aligning with our findings in Section 4. NoPE and ALiBi effectively generalize to larger context sizes up to $1800$. However, as context sizes grow beyond that, ALiBi exhibits relative stability compared to NoPE, though both models show a pattern of increasing perplexity, potentially indicating inefficiencies to capture longer dependencies.

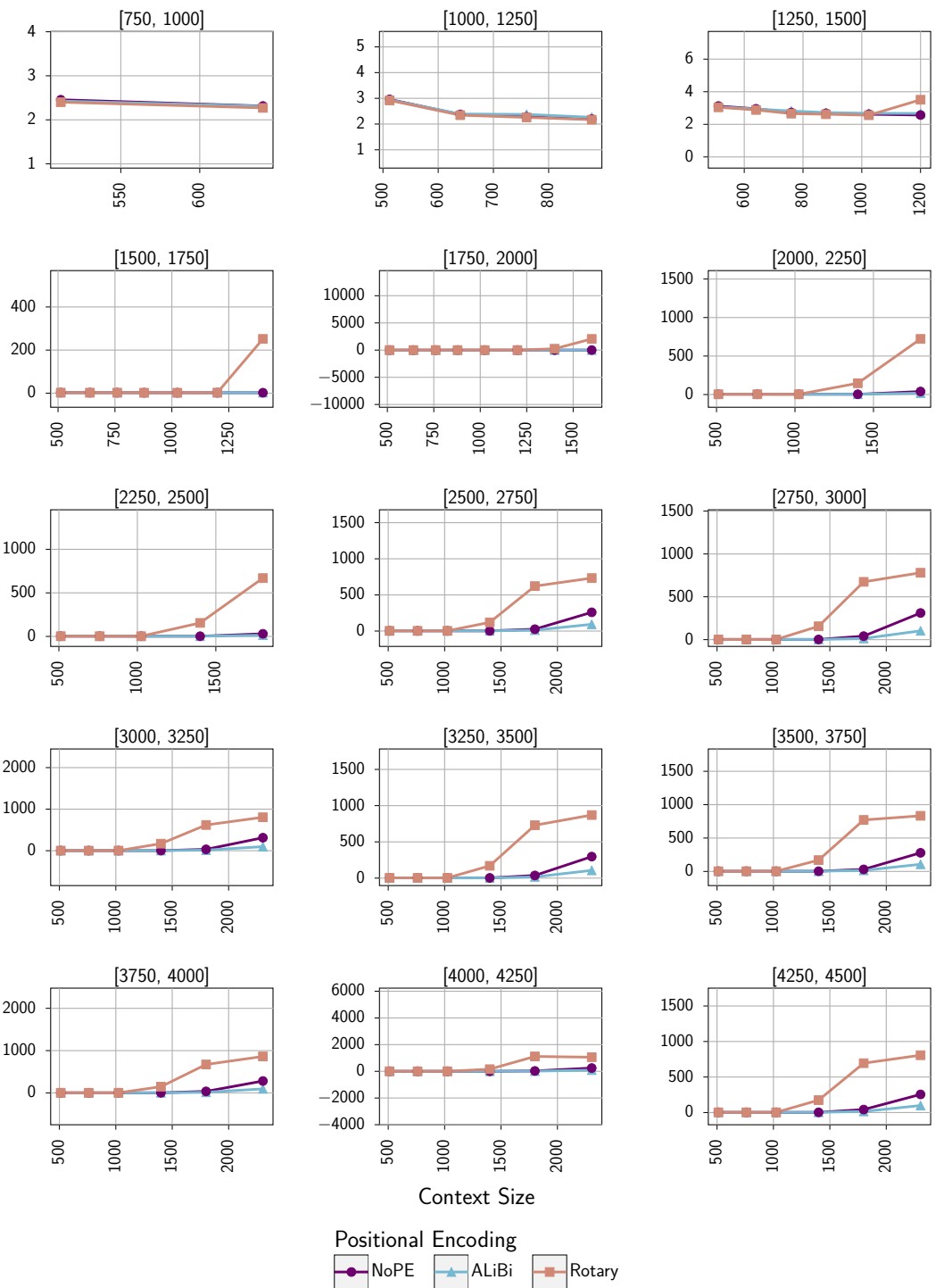

Figure F.2: Perplexity of the three pretrained models on examples from various length buckets. Each model is trained with different positional encoding schemes on identical training data and hyperparameters.

Table 3: The detailed perplexity (represented along with Figure F.2 for completeness). of the three pretrained models on examples from various length buckets (Part I)

| Model | Context Size | | | | | | | | | | | |
|---|---|---|---|---|---|---|---|---|---|---|---|---|
| **Length Bucket [750, 1000]** | | | | | | | | | | | | |
| | 512 | 640 | 760 | 878 | 1024 | 1200 | 1400 | 1600 | 1800 | 2048 | 2304 | 2560 |
| NoPE | 2.456 | 2.314 | - | - | - | - | - | - | - | - | - | - |
| ALiBi | 2.427 | 2.310 | - | - | - | - | - | - | - | - | - | - |
| Rotary | 2.401 | 2.271 | - | - | - | - | - | - | - | - | - | - |
| **Length Bucket [1000, 1250]** | | | | | | | | | | | | |
| | 512 | 640 | 760 | 878 | 1024 | 1200 | 1400 | 1600 | 1800 | 2048 | 2304 | 2560 |
| NoPE | 2.958 | 2.372 | 2.282 | 2.192 | - | - | - | - | - | - | - | - |
| ALiBi | 2.944 | 2.392 | 2.375 | 2.263 | - | - | - | - | - | - | - | - |
| Rotary | 2.920 | 2.344 | 2.259 | 2.170 | - | - | - | - | - | - | - | - |
| **Length Bucket [1250, 1500]** | | | | | | | | | | | | |
| | 512 | 640 | 760 | 878 | 1024 | 1200 | 1400 | 1600 | 1800 | 2048 | 2304 | 2560 |
| NoPE | 3.122 | 2.951 | 2.722 | 2.676 | 2.618 | 2.570 | - | - | - | - | - | - |
| ALiBi | 3.077 | 2.940 | 2.816 | 2.723 | 2.678 | 2.670 | - | - | - | - | - | - |
| Rotary | 3.044 | 2.882 | 2.657 | 2.619 | 2.559 | 3.508 | - | - | - | - | - | - |
| **Length Bucket [1500, 1750]** | | | | | | | | | | | | |
| | 512 | 640 | 760 | 878 | 1024 | 1200 | 1400 | 1600 | 1800 | 2048 | 2304 | 2560 |
| NoPE | 2.620 | 2.541 | 2.467 | 2.258 | 2.187 | 2.158 | 2.121 | - | - | - | - | - |
| ALiBi | 2.616 | 2.564 | 2.527 | 2.305 | 2.226 | 2.249 | 2.307 | - | - | - | - | - |
| Rotary | 2.580 | 2.502 | 2.418 | 2.220 | 2.146 | 2.749 | 251.684 | - | - | - | - | - |
| **Length Bucket [1750, 2000]** | | | | | | | | | | | | |
| | 512 | 640 | 760 | 878 | 1024 | 1200 | 1400 | 1600 | 1800 | 2048 | 2304 | 2560 |
| NoPE | 3.002 | 2.911 | 2.853 | 2.704 | 2.436 | 2.397 | 2.399 | 4.581 | - | - | - | - |
| ALiBi | 2.966 | 2.908 | 2.954 | 2.774 | 2.480 | 2.487 | 2.595 | 4.114 | - | - | - | - |
| Rotary | 2.929 | 2.845 | 2.790 | 2.646 | 2.390 | 3.098 | 248.698 | 2054.099 | - | - | - | - |
| **Length Bucket [2000, 2250]** | | | | | | | | | | | | |
| | 512 | 640 | 760 | 878 | 1024 | 1200 | 1400 | 1600 | 1800 | 2048 | 2304 | 2560 |
| NoPE | 4.130 | 3.998 | 3.892 | 3.849 | 3.751 | 3.348 | 3.338 | 6.325 | 39.413 | - | - | - |
| ALiBi | 4.087 | 3.988 | 4.049 | 3.983 | 3.823 | 3.479 | 3.616 | 5.365 | 18.731 | - | - | - |
| Rotary | 4.007 | 3.878 | 3.787 | 3.744 | 3.657 | 4.190 | 146.958 | 531.824 | 722.276 | - | - | - |
| **Length Bucket [2250, 2500]** | | | | | | | | | | | | |
| | 512 | 640 | 760 | 878 | 1024 | 1200 | 1400 | 1600 | 1800 | 2048 | 2304 | 2560 |
| NoPE | 2.778 | 2.642 | 2.555 | 2.493 | 2.450 | 2.321 | 2.159 | 2.859 | 30.012 | 146.219 | - | - |
| ALiBi | 2.764 | 2.655 | 2.647 | 2.570 | 2.499 | 2.441 | 2.361 | 3.695 | 14.173 | 44.089 | - | - |
| Rotary | 2.732 | 2.600 | 2.514 | 2.448 | 2.415 | 2.864 | 155.077 | 532.601 | 669.556 | 750.022 | - | - |
| **Length Bucket [2500, 2750]** | | | | | | | | | | | | |
| | 512 | 640 | 760 | 878 | 1024 | 1200 | 1400 | 1600 | 1800 | 2048 | 2304 | 2560 |
| NoPE | 2.805 | 2.716 | 2.645 | 2.614 | 2.574 | 2.534 | 2.367 | 2.954 | 25.694 | 114.870 | 259.222 | - |
| ALiBi | 2.801 | 2.732 | 2.721 | 2.726 | 2.651 | 2.681 | 2.575 | 3.585 | 12.350 | 37.746 | 93.094 | - |
| Rotary | 2.776 | 2.686 | 2.615 | 2.587 | 2.554 | 3.302 | 120.402 | 503.155 | 621.873 | 676.160 | 732.808 | - |
| **Length Bucket [2750, 3000]** | | | | | | | | | | | | |
| | 512 | 640 | 760 | 878 | 1024 | 1200 | 1400 | 1600 | 1800 | 2048 | 2304 | 2560 |
| NoPE | 2.850 | 2.776 | 2.728 | 2.694 | 2.666 | 2.620 | 2.504 | 3.839 | 41.194 | 151.810 | 311.138 | 536.225 |
| ALiBi | 2.830 | 2.789 | 2.825 | 2.766 | 2.735 | 2.742 | 2.741 | 3.680 | 13.814 | 42.951 | 102.930 | 150.322 |
| Rotary | 2.774 | 2.702 | 2.656 | 2.629 | 2.607 | 3.323 | 156.282 | 506.789 | 674.506 | 676.882 | 780.594 | 760.104 |
| **Length Bucket [3000, 3250]** | | | | | | | | | | | | |
| | 512 | 640 | 760 | 878 | 1024 | 1200 | 1400 | 1600 | 1800 | 2048 | 2304 | 2560 |
| NoPE | 2.783 | 2.697 | 2.639 | 2.595 | 2.555 | 2.519 | 2.498 | 4.239 | 32.277 | 153.248 | 312.356 | 521.752 |
| ALiBi | 2.782 | 2.746 | 2.750 | 2.688 | 2.665 | 2.692 | 2.763 | 3.847 | 13.542 | 41.208 | 97.685 | 143.644 |
| Rotary | 2.733 | 2.649 | 2.589 | 2.542 | 2.506 | 3.107 | 169.800 | 471.389 | 619.115 | 670.209 | 804.750 | 824.417 |
| **Length Bucket [3250, 3500]** | | | | | | | | | | | | |
| | 512 | 640 | 760 | 878 | 1024 | 1200 | 1400 | 1600 | 1800 | 2048 | 2304 | 2560 |
| NoPE | 3.260 | 3.140 | 3.061 | 3.009 | 2.878 | 2.828 | 2.866 | 6.048 | 36.581 | 143.613 | 296.817 | 479.934 |
| ALiBi | 3.206 | 3.116 | 3.127 | 3.073 | 2.922 | 2.940 | 3.104 | 5.077 | 16.838 | 47.832 | 106.607 | 151.288 |
| Rotary | 3.181 | 3.061 | 2.985 | 2.935 | 2.823 | 3.525 | 169.851 | 664.657 | 729.844 | 861.122 | 870.443 | 943.464 |

Table 4: The detailed perplexity (represented along with Figure F.2 for completeness). of the three pretrained models on examples from various length buckets (Part II)

| Model | Context Size | | | | | | | | | | | |
|-------|------|------|------|------|------|------|------|------|------|------|------|------|

Length Bucket [3500, 3750]

| | 512 | 640 | 760 | 878 | 1024 | 1200 | 1400 | 1600 | 1800 | 2048 | 2304 | 2560 |
|-------|------|------|------|------|------|------|------|------|------|------|------|------|
| NoPE | 3.070 | 3.003 | 2.969 | 2.942 | 2.894 | 2.826 | 2.816 | 3.808 | 30.840 | 122.338 | 278.935 | 473.120 |
| ALiBi | 3.054 | 3.011 | 3.069 | 3.006 | 2.954 | 2.971 | 3.107 | 4.845 | 15.691 | 46.406 | 106.966 | 156.849 |
| Rotary | 3.020 | 2.958 | 2.923 | 2.892 | 2.844 | 3.719 | 169.445 | 604.442 | 771.338 | 766.585 | 833.169 | 888.627 |

Length Bucket [3750, 4000]

| | 512 | 640 | 760 | 878 | 1024 | 1200 | 1400 | 1600 | 1800 | 2048 | 2304 | 2560 |
|-------|------|------|------|------|------|------|------|------|------|------|------|------|
| NoPE | 2.703 | 2.593 | 2.564 | 2.517 | 2.466 | 2.440 | 2.427 | 3.447 | 35.851 | 135.291 | 281.145 | 438.679 |
| ALiBi | 2.657 | 2.570 | 2.614 | 2.553 | 2.515 | 2.541 | 2.639 | 4.049 | 14.112 | 40.070 | 94.953 | 138.232 |
| Rotary | 2.645 | 2.536 | 2.506 | 2.461 | 2.416 | 2.981 | 147.012 | 508.386 | 671.961 | 707.306 | 861.924 | 873.948 |

Length Bucket [4000, 4250]

| | 512 | 640 | 760 | 878 | 1024 | 1200 | 1400 | 1600 | 1800 | 2048 | 2304 | 2560 |
|-------|------|------|------|------|------|------|------|------|------|------|------|------|
| NoPE | 3.096 | 3.021 | 2.947 | 2.905 | 2.862 | 2.843 | 2.809 | 3.459 | 27.171 | 109.982 | 244.809 | 414.226 |
| ALiBi | 3.086 | 3.040 | 3.064 | 3.028 | 2.931 | 2.983 | 3.058 | 4.316 | 13.509 | 37.900 | 88.864 | 129.419 |
| Rotary | 3.054 | 2.973 | 2.906 | 2.868 | 2.831 | 3.626 | 158.974 | 723.175 | 1114.522 | 989.245 | 1057.793 | 960.442 |

Length Bucket [4250, 4500]

| | 512 | 640 | 760 | 878 | 1024 | 1200 | 1400 | 1600 | 1800 | 2048 | 2304 | 2560 |
|-------|------|------|------|------|------|------|------|------|------|------|------|------|
| NoPE | 3.056 | 2.992 | 2.921 | 2.869 | 2.815 | 2.786 | 2.872 | 8.839 | 42.054 | 133.110 | 255.307 | 418.937 |
| ALiBi | 3.004 | 3.008 | 3.022 | 2.975 | 2.913 | 2.975 | 3.097 | 4.913 | 15.522 | 43.614 | 98.910 | 142.342 |
| Rotary | 2.958 | 2.899 | 2.838 | 2.775 | 2.731 | 3.603 | 174.717 | 528.865 | 694.649 | 741.368 | 806.988 | 857.006 |

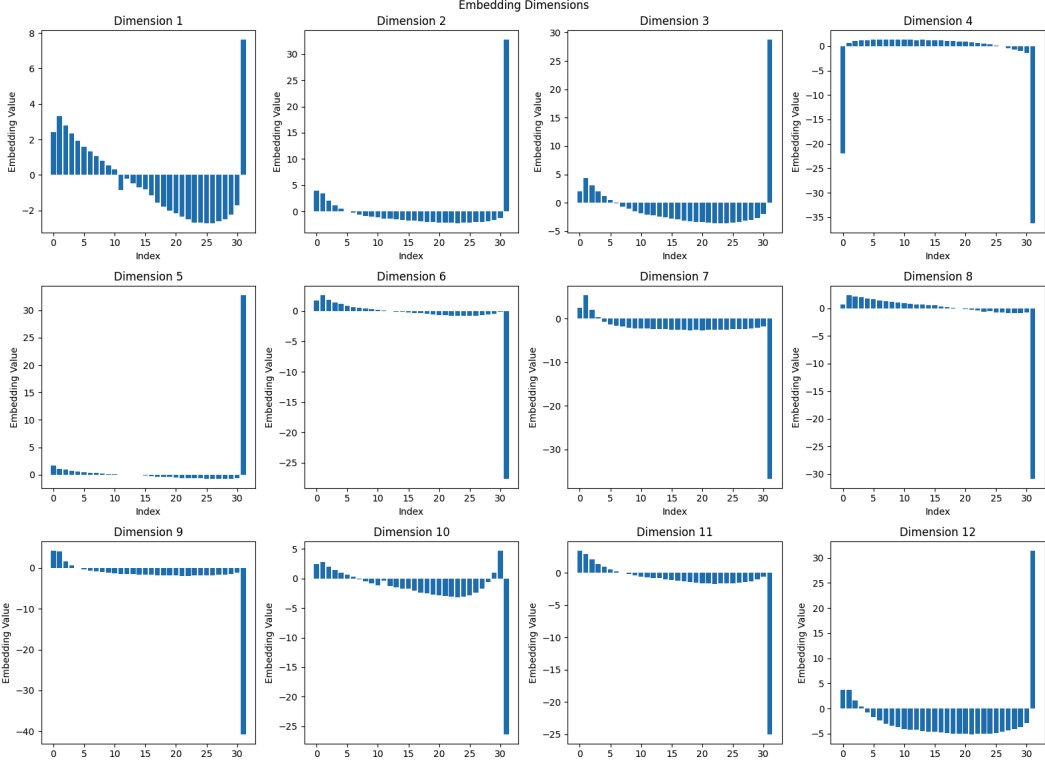

Figure F.3: The learnt bias value $b_{\text{bucket}(.)}$ (taken from the pretrained T5-3b model) visualized for relative distance bucket $\text{bucket}(.) \in [0, 31]$) and 12 heads. Bucket 31 is exclusively used for all distances larger than 128 tokens. These results demonstrate that while T5 has heads that mask distant tokens, other heads attend to distant tokens when required.

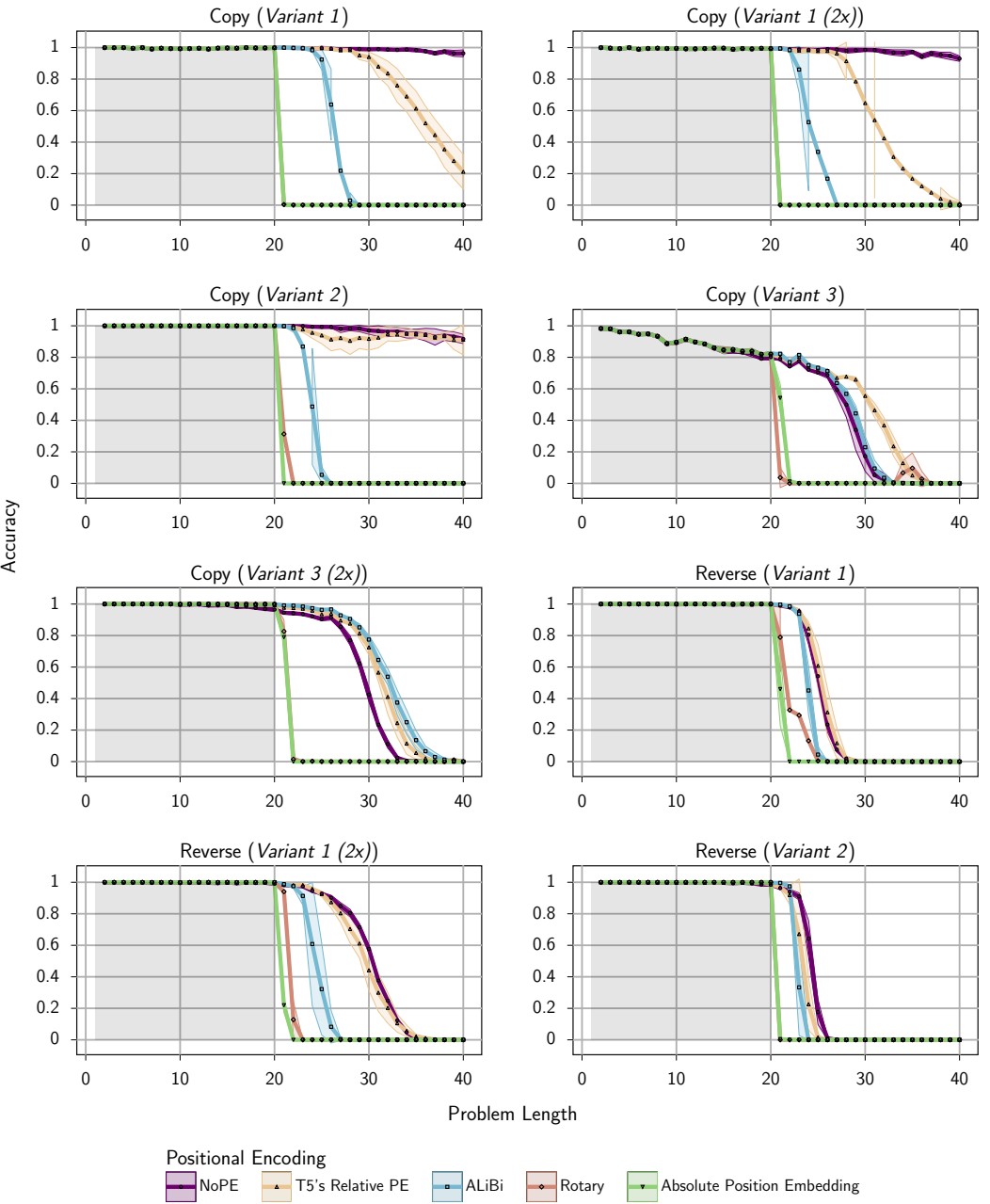

Figure F.4: Generalization behavior of positional encoding schemes on Primitive tasks.

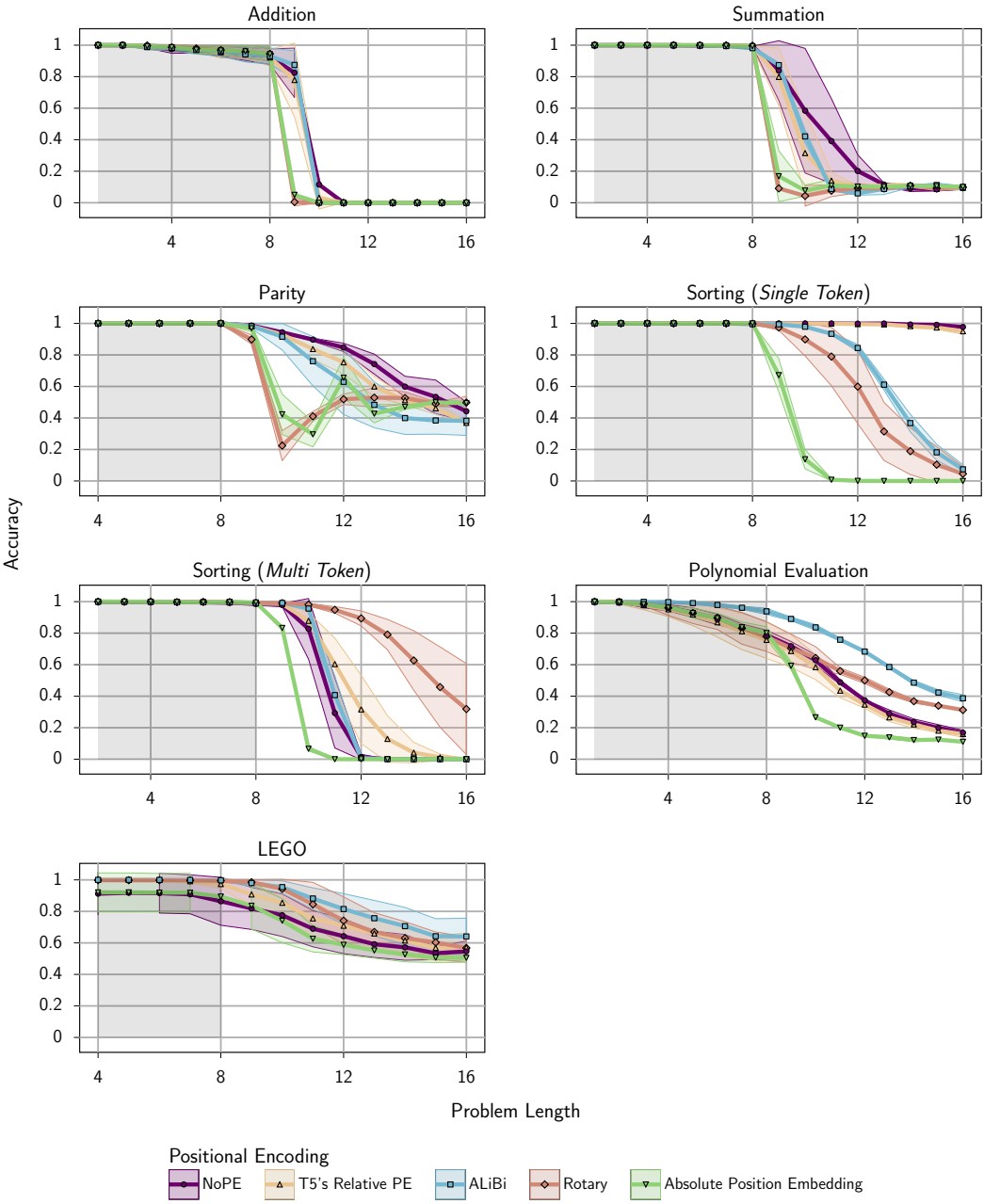

Figure F.5: Generalization behavior of positional encoding schemes on Mathematical & Reasoning tasks.

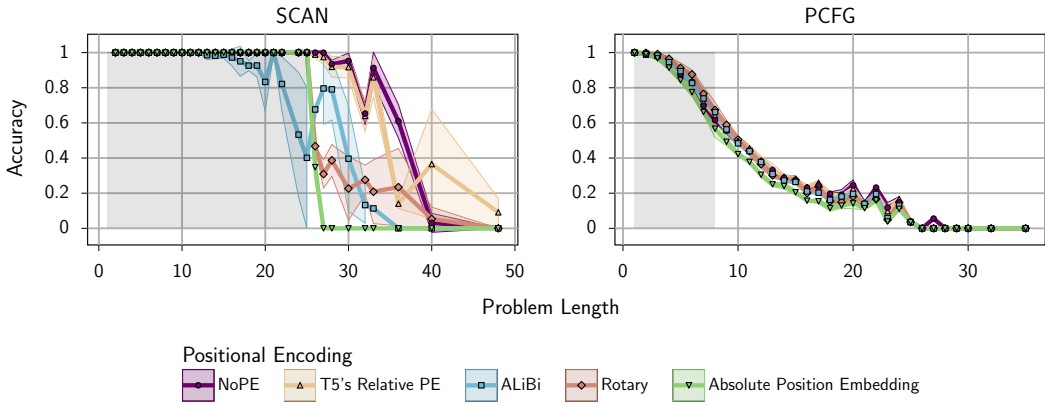

Figure F.6: Generalization behavior of positional encoding schemes on Classic Length Generalization tasks.

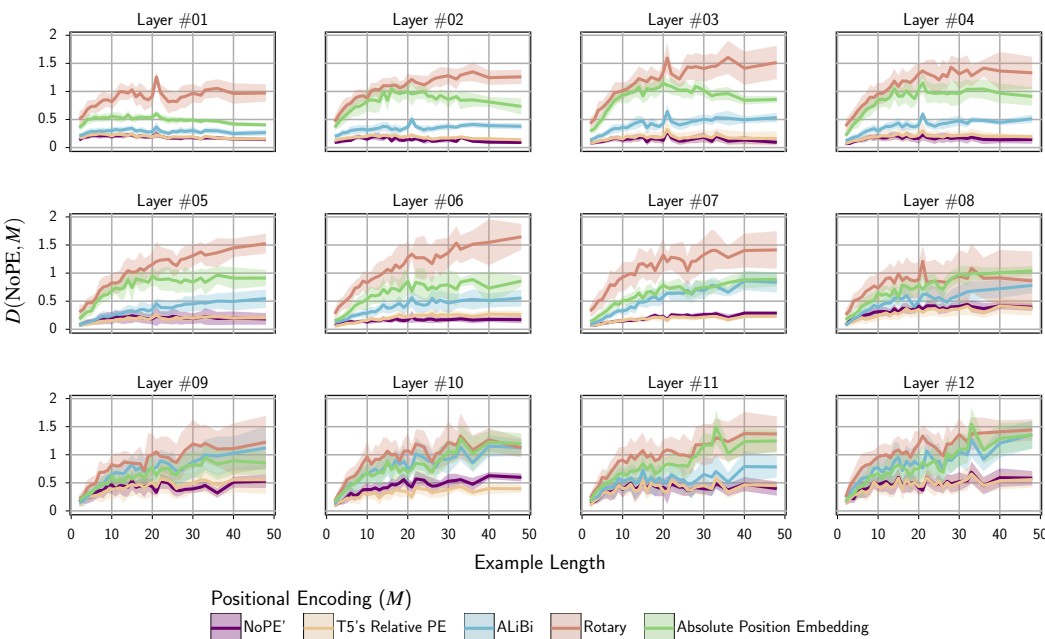

Figure F.7: Layer-wise distance of No PE's attention patterns with other positional encoding schemes measured across instances of SCAN dataset.

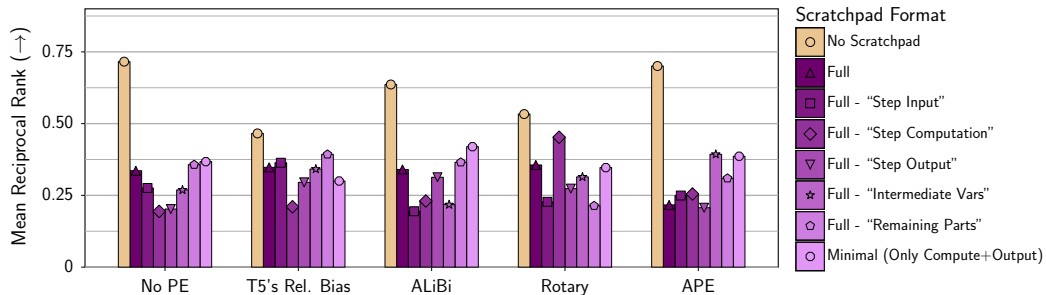

Figure F.8: The optimal scratchpad format is aggregated across all datasets per each model. The optimal scratchpad format is different for each model.

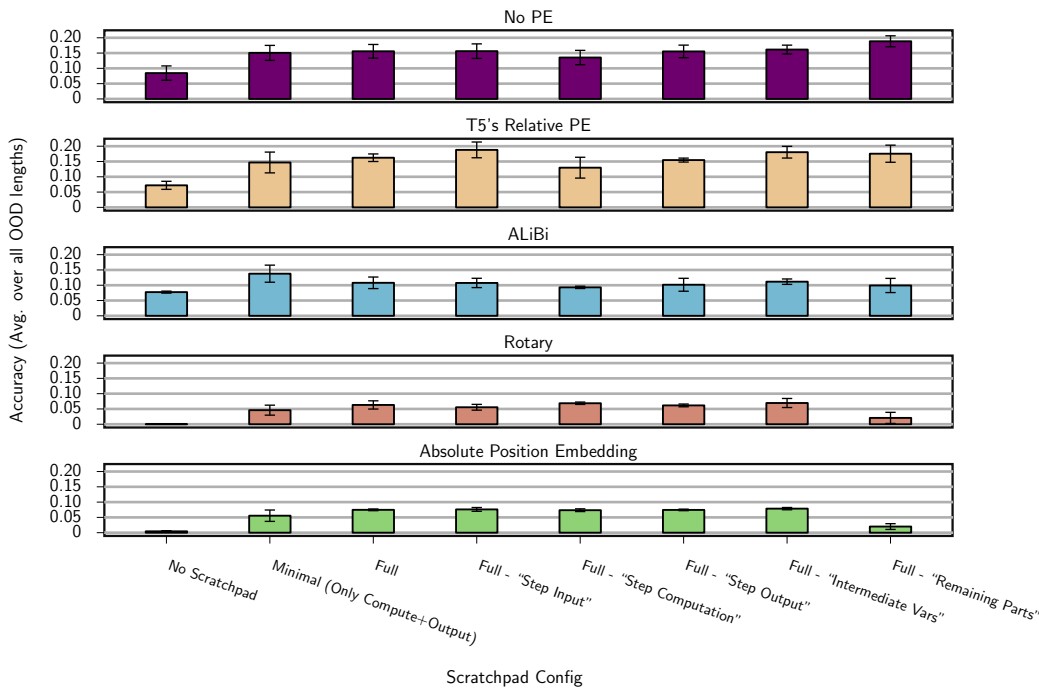

Figure F.9: Generalization of various scratchpad formats for each model on the **Addition** task.

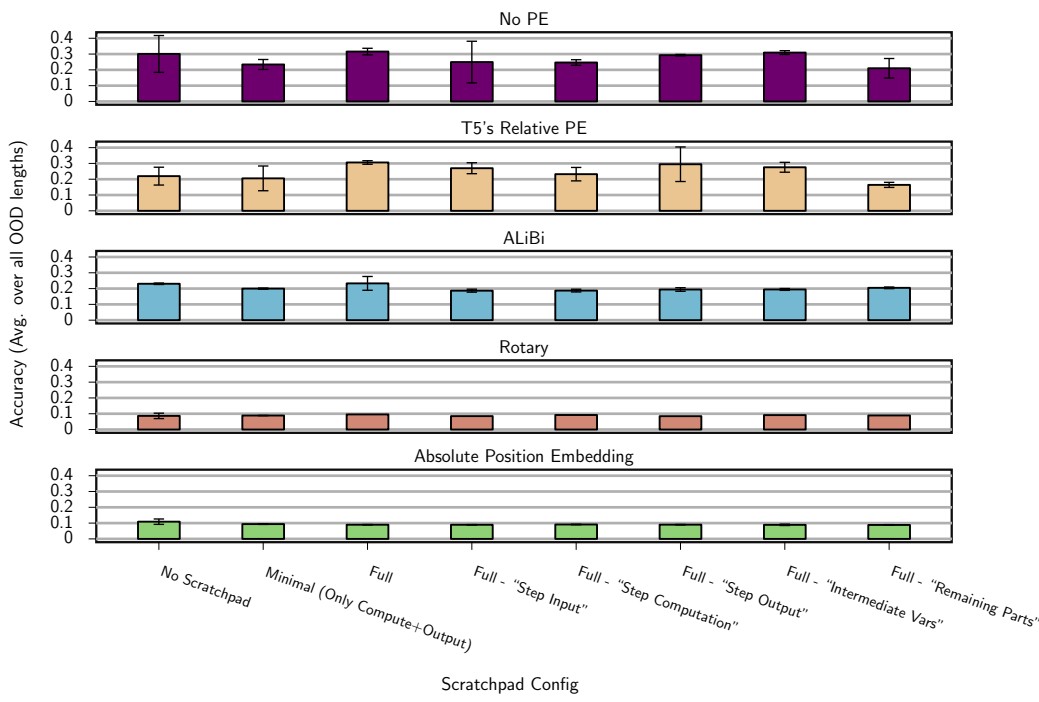

Figure F.10: Generalization of various scratchpad formats for each model on the **Summation** task.

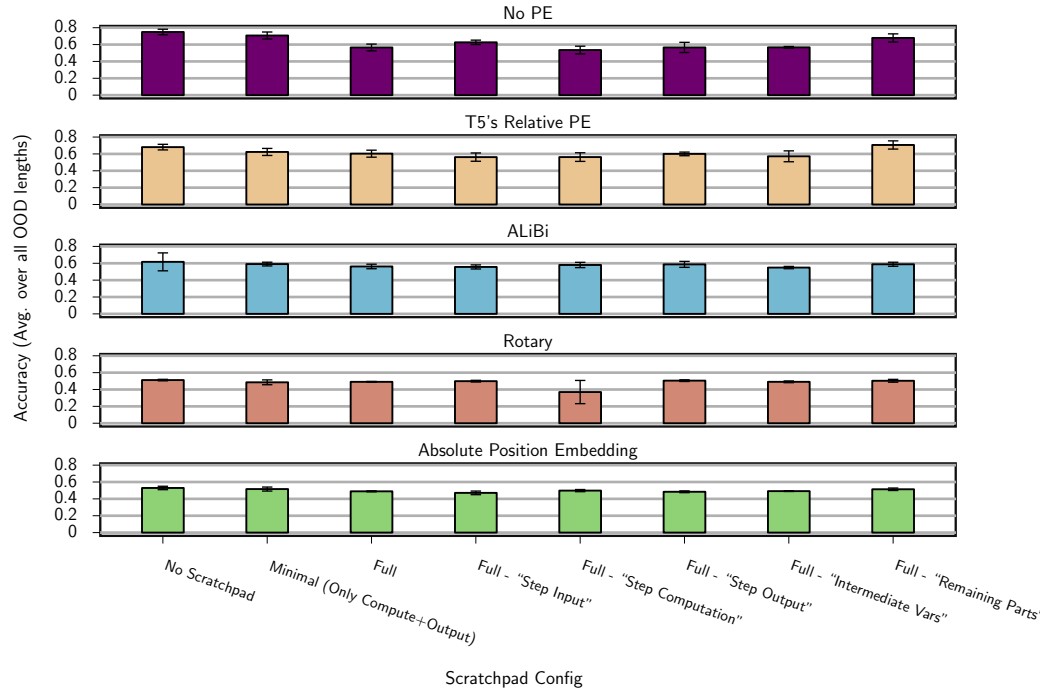

Figure F.11: Generalization of various scratchpad formats for each model on the **Parity** task.

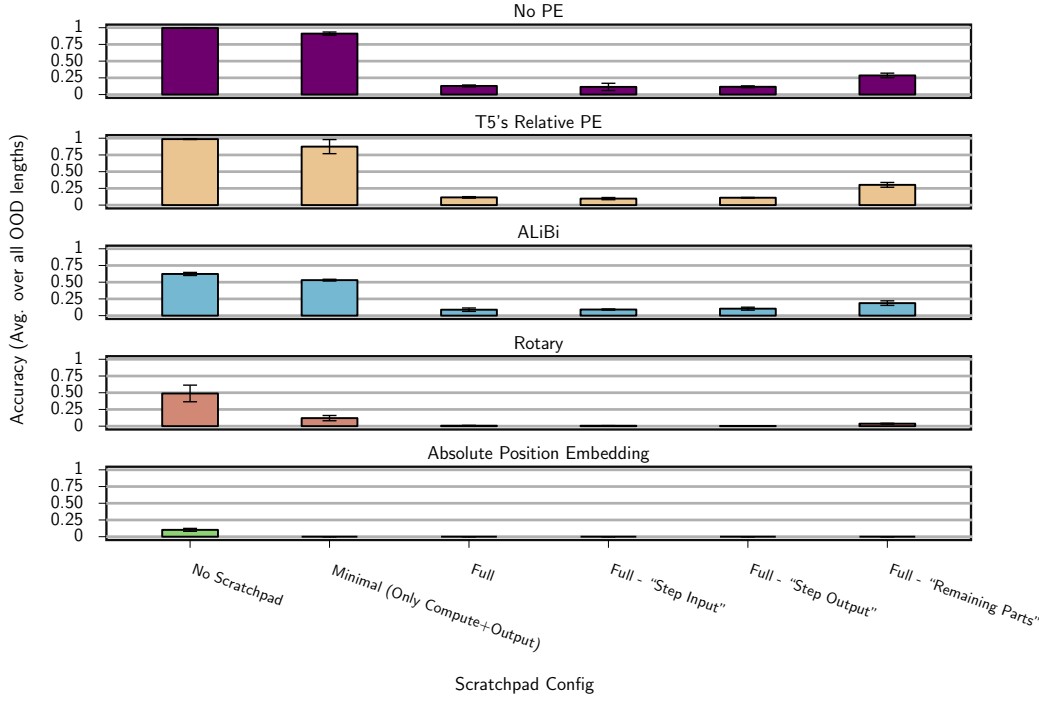

Figure F.12: Generalization of various scratchpad formats for each model on the **Sorting** task (Single Digit).

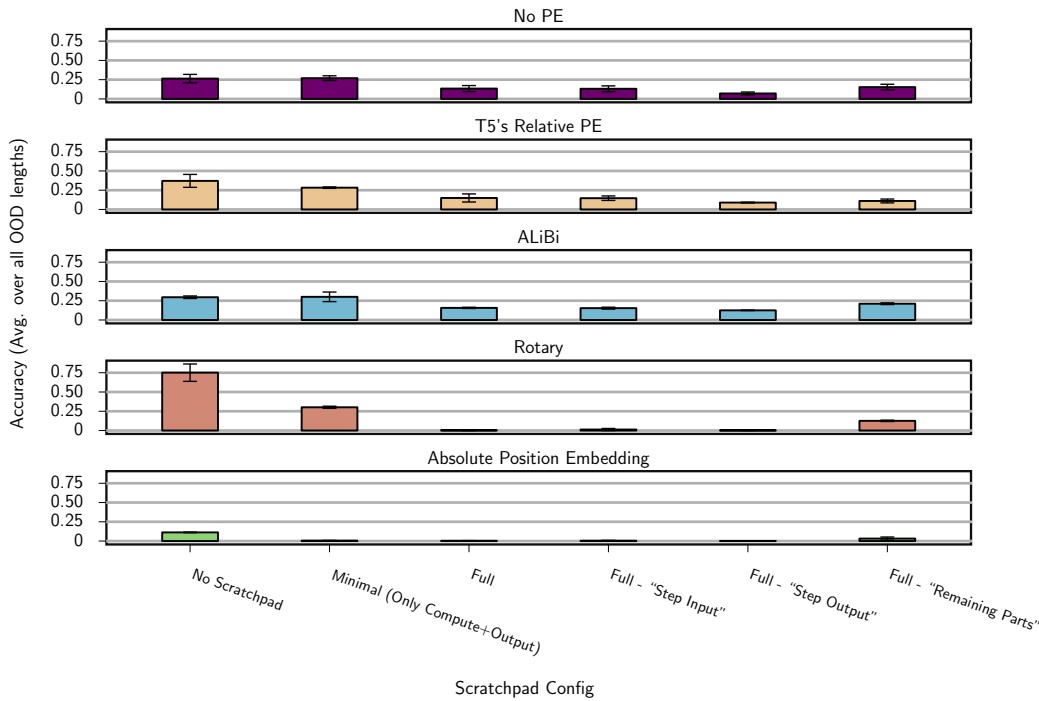

Figure F.13: Generalization of various scratchpad formats for each model on the **Sorting** task (Multi Digit).

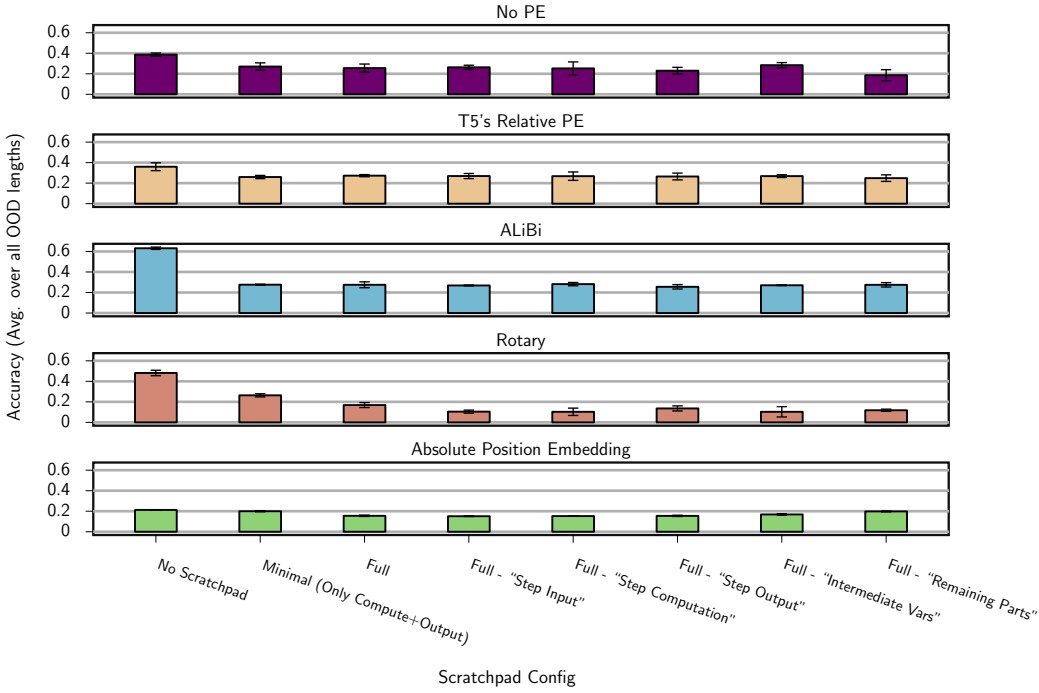

Figure F.14: Generalization of various scratchpad formats for each model on the **Polynomial Evaluation** task.

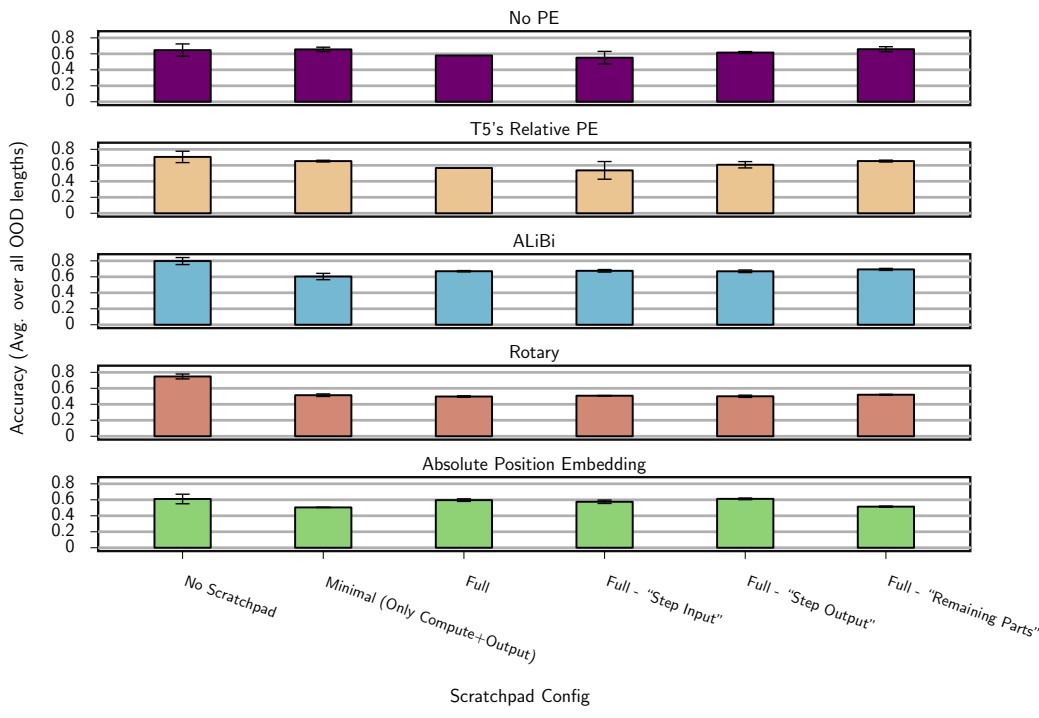

Figure F.15: Generalization of various scratchpad formats for each model on the **LEGO** task.

