# OpenReview forum: "The Impact of Positional Encoding on Length Generalization in Transformers"
_NeurIPS.cc/2023/Conference — NeurIPS 2023 poster_

### Official Review · Reviewer_C4oP · 2023-07-06

**Soundness:** 3 good
**Presentation:** 3 good
**Contribution:** 2 fair
**Rating:** 5
**Confidence:** 4

**Summary:**

Beyond model training, sequence length is one of the crucial aspect impacting the fluency, accuracy and completeness of LM outputs. For transformer based model's positional embeddings (PE) are used to explicitly represent the location of tokens. Over the past few years several works have attempted to propose a better PE approach, particularly aiming to improve over the original absolute PE of the Transformer.

This work, proposes a no positional encoding (noPE) model training for decoder only Transformer, to better generalize on generation length. The proposed noPE LM is evaluated on set of maths and reasoning tasks, using a small size (~100M parameter) LM. Findings show that noPE based LM performs similarly with relative PE based model (from the T5 model), and outperforms models with alternative PE variants such as Rotary, AliBi, and absolute PE for length generalization. Along with the evaluation results, detailed empirical analysis and theoretical arguments are provided for why noPE LM can intrinsically support an absolute and relative token position representations.


**Strengths:**

This work, provided an in-depth analysis of Transformer LM training and inference without using PE. Although, the focus is to address the question of LM length generalization, the findings inform future possibility of an increasingly simplified transformer based LM's. In addition to the surprising results where noPE works in a similar way as the best performing relative PE (T5's PE), the work, delivered a thorough discussion as to why noPE works.



**Weaknesses:**

Despite the finding not using PE (noPE) can generalize for sequence length, the following points remain unanswered or not fully investigated:
- Tasks selected for modeling and evaluation does not cover challenging length generalization scenarios. For instance, the noPE approach could have been testing using a summarization task which is highly impacted by the length of the summary.
- Regardless of the provided results on length generalization, detailed and task specific performance metrics comparing LM's with PE's and noPE LM is missing. The absence of these comparison leaves open question on task specific performance comparison.

**Questions:**

- In addition to the mean rank, can authors add more task specific evaluation metrics, or could you please provide more motivation on why not to report task specific performance metric?
- This can be a hypothetical one, have you considered a fine-tuning approach for LM's, for instance removing the PE when fine-tuning for down-stream tasks?
- Can authors expand more on the selection of the LM size (100M parameters)? can we expect similar findings to hold if model size changes (i.e. larger LM's with more diverse and large scale train data)?

**Limitations:**

Limitation is provided, specifically explaining why the proposed noPE is not applied for large scale LM training. The limitation, not being able to access publicly available LM can be further expanded, for instance why not use 500M? If there is any identified societal impact its encouraged to include it.

---

> ### Author Rebuttal · Authors · 2023-08-09
>
> We appreciate the reviewer's insightful feedback and questions, and we will now address them:
>
> > Inclusion of other tasks such as summarization
>
> We acknowledge the reviewer's concern about including more challenging tasks, such as summarization. However, we'd like to clarify why we focused on reasoning and mathematical tasks in our evaluation.
>
> 1. **Standard Test Bed**:
> These tasks have become the de facto standard for assessing length generalization in related literature [1,2,3,4]. and although they may seem simple, they require compositionality skills to solve, fundamental for more complex tasks.
> 2. **Control over Problem Length**: The synthetic nature of the chosen tasks enables full control over the data generation process. It permits a train/test split designed to measure length generalization exclusively, with task difficulty tied to problem length.
> 3. **Challenges with Naturally Collected Tasks**: While tasks like summarization might appear more ideal, creating a train/test split measuring length generalization is highly non-trivial for such naturally collected/crowdsourced tasks. Notably, sample length and sample difficulty are not necessarily correlated.  One might be tempted to create splits using sample length, however, since we don’t have control over the data generating distribution, longer samples may not be necessarily more challenging to solve. For example, it is unclear if testing with longer documents or summaries would reflect increased difficulty. Longer documents might not be more difficult (e.g. source document mostly contains irrelevant content), and longer summaries may not entail harder examples (e.g source document is so small that creating a summary only requires a mere copy-paste from the input).
> 4. **Alignment with Prior Work**: Considering these factors, we followed existing practices, evaluating on a similar set of tasks and including a diverse range (10 in our work) to offer a comprehensive view of the results.
>
> In conclusion, our task selection was grounded in established methodologies for assessing length generalization. We believe this approach provided a robust examination of the subject, though we appreciate any suggestion or elaboration on our choice of tasks.
>
>
> > Task-specific performance metrics
>
> We appreciate the reviewer’s comment. To address this concern, we would like to clarify and politely remind the Reviewer that our evaluation **does indeed include task-specific metrics**, particularly exact-match accuracy, which are reported not only per each task but also for each length bucket in both the main paper and the appendix section.
>
> Particularly, Fig. 3 in the main paper illustrates six task-specific evaluations, encompassing both in-distribution (IID) and out-of-distribution (OOD) scenarios. Additionally, Figs. E.2, E.3, and E.4 in the appendix provide detailed evaluations for each specific task. The evaluation result for all mathematical tasks, complete with eight different scratchpad format configurations, are thoroughly documented in Figs. E.7 to E.13.
>
> Additionally, the motivation for mean rank is to offer a more holistic perspective. Utilizing mean rank is a standard practice [5] in situations where the evaluation spans multiple diverse tasks in which the mean rank and task-specific results are reported simultaneously. We directly follow this practice as it provides a more comprehensive view.
>
> We hope this clarification reassures you of the rigor and depth of our evaluation (as acknowledged by Reviewer rWd9). If any part still appears ambiguous or incomplete, we are open to further elaboration to enhance the clarity of our work.
>
> > Finetuning approach for removing PE
>
> We are glad our thought processes match. At the preliminary stage of our research, we did indeed consider removing positional encoding in pretrained models and subsequently fine-tuning them on downstream tasks. This consideration, however, was tempered by existing findings in the literature. Specifically, prior work by Sinha et al. [6] has demonstrated that positional encoding instills such a bias during pretraining that proves challenging to mitigate, even with further finetuning. In light of these observations, we made the deliberate choice to forgo fine-tuning to avoid introducing a potential confounder into our study. Nevertheless, we recognize the intriguing nature of this question and fully endorse its exploration as an important avenue for future research.
>
>
> > Selection of the model size and transferability to larger models
>
> We appreciate the question about the size of our models. Without LLMs trained with different PEs under uniform conditions, assessing the effect of PEs on length in pretrained or bigger models is tough. As a result, we had to train decoder-only Transformer models from scratch, constrained by our computational limits. The model size chosen aligns with recent research on length generalization [1,2].
>
> Nonetheless, despite our limited budget, we've begun addressing concerns by including initial scaling and language model pretraining results in the rebuttal period (see General Response). These insights show patterns aligned with our findings, suggesting our conclusions may apply at larger scales. We're gathering resources to conduct pretraining at a 1B-parameter level, albeit limited.
>
> We express our gratitude once more to the reviewer for their valuable time and insights. We hope that our detailed explanations have resolved any remaining questions. Would you consider increasing your scores given the points of clarification that we have provided above?
> - [1] Delétang et al, https://arxiv.org/abs/2207.02098, ICLR 2023
> - [2] Ruoss et al,https://aclanthology.org/2023.acl-short.161.pdf, ACL 2023
> - [3] Anil et al, https://arxiv.org/abs/2207.04901, NeurIPS 2022
> - [4] Ontañón et al, https://arxiv.org/abs/2108.04378, ACL 2022
> - [5] Liang et al, https://arxiv.org/abs/2211.09110, 2022.
> - [6] Sinha et al, ​​https://arxiv.org/abs/2210.12574, EMNLP 2022 Findings

---

> > ### Author Response · Authors · 2023-08-18
> >
> > Dear Reviewer,
> >
> > We appreciate the effort you’ve put into reviewing our paper and offering insightful feedback. We’re wondering if you’ve had a chance to go through our response.  As the discussion period is nearly over, we’d love to address any lingering queries or concerns you may have.

---

### Official Review · Reviewer_rWd9 · 2023-07-06

**Soundness:** 3 good
**Presentation:** 3 good
**Contribution:** 3 good
**Rating:** 7
**Confidence:** 4

**Summary:**

This work follows a long line of work analysing the effectiveness of various types of positional encodings and, in particular, a line of work exploring the effectiveness of causal masking without any positional encodings [1 - 8]. To that end, they evaluate the length generalisation performance of Transformers trained with various types of positional encoding schemes on a diverse set of reasoning and algorithmic tasks and find that Transformers without positional encoding schemes generalise better to higher lengths than other encoding schemes.


They consider 3 three types of tasks -- primitive (copying or reversing strings), algorithmic (addition, parity, etc.), and prior length generalisation benchmarks (e.g. SCAN). They evaluate five different encoding schemes, including Alibi and rotary encodings. They use autoregressive decoder-only Transformers of sizes of the order of 100M. For each task, Transformers are first trained on a set of examples using a language modelling objective and are then evaluated on examples of higher lengths.

Across the three types of tasks, they find that Transformers without positional encoding (but with masking) and T5's relative bias outperform other encoding schemes. Additionally, they show via a construction that Transformers without positional encodings can simulate both absolute and relative encoding schemes. They compare the attention patterns of Transformers trained without positional encoding schemes with other schemes and find that the attention patterns are most similar to relative positional encodings. Lastly, they find that when trained with a scratchpad, T5's relative positional encoding scheme and no positional encodings seem more effective than others.



**Strengths:**


(S1) **Interesting results.** Overall, I believe there are several interesting findings in the paper. I believe the contributions in the paper take a step forward in understanding the effectiveness of Transformers with just causal masking. The analysis with the scratchpad setup demonstrating the importance of PE for scratchpad and the effect of scratchpad for length generalisation is interesting as well.


(S2) **Sound experimental setup and well-motivated.** The question of understanding and identifying the most effective positional encoding scheme is an important problem, and this work takes a step forward in that direction. The choice of tasks seems interesting, and the evaluation with Transformers trained with language modelling objectives is more aligned with practice than prior works that have explored this direction.


(S3) **Well written.** The paper is well-written, the motivations are clearly outlined, and the arguments are well-presented.



**Weaknesses:**


(W1) **Novelty of results is slightly overclaimed.** Based on my limited knowledge, the use of Transformers without positional embeddings and with only causal masking was first studied in [1]. It was later analysed empirically [2,3] and theoretically [3, 5] in further works. These works mostly suggested that Transformers without positional encodings are competitive with standard encoding schemes on machine translation and language modelling tasks. On synthetic formal language tasks, [4] and more recently [6] have noted that using no positional encoding leads to better length generalisation performance. In more practical settings, recently [7, 8] have shown that large language models trained with only causal masking are competitive with models with positional encoding.

The fact that Transformers with only causal masking can encode positional information and generalise well to higher lengths is not entirely new. To be fair to the authors, I do not think that they explicitly claim this. I think the paper does a good job of establishing that just causal masking is better at length generalisation than various other schemes in a more systematic way. However, given the prior work exploring Transformers with only causal masking, the framing of the introduction seems a bit odd, and some of the papers mentioned above [1-8] are mentioned passively in the paper. It could be helpful to readers unfamiliar with the area if the prior work was presented in a more comprehensive manner.


(W2) **Effect of scale.** Since the tasks in the paper are confined to synthetic tasks and models with ~100M parameters, it is still unclear if training LLMs without positional encodings will be better for length generalisation in practice. Modifications to Transformers at this scale for effectiveness on such tasks have often not scaled to LLMs in practice. As mentioned in the paper, that could require a larger computational budget which is understandable to some extent. But at the same time, comparing Transformers trained for language modelling on text data on higher lengths would have been useful. Based on the results in [7, 8], it seems like positional encoding schemes help improve perplexity in language modelling, but it may be interesting to see if that changes for length generalisation tasks.






[1] DiSAN: Directional Self-Attention Network for RNN/CNN-Free Language Understanding. 2017
[2] Assessing the Ability of Self-Attention Networks to Learn Word Order. 2019
[3] Transformer Dissection: An Unified Understanding for Transformer's Attention via the Lens of Kernel. 2019
[4] On the Ability and Limitations of Transformers to Recognise Formal Languages. 2020
[5] On the Computational Power of Transformers and its Implications in Sequence Modeling. 2020
[6] Transformers Learn Shortcuts to Automata. 2022
[7] Transformer Language Models without Positional Encodings Still Learn Positional Information. 2022
[8] Bloom: A parameter open-access multilingual language model. 2022






**Questions:**

(Q1) Regarding Section 5.2, if the models are trained with different positional encoding schemes from scratch, how can similarity between attention patterns help us determine how the model is computing positional information? Is it not possible that the models have very different learning trajectories? What about cases when Transformer without positional encoding performs better than relative encodings, wouldn't the underlying mechanism be different in that scenario? I think it is interesting the attention patterns without encodings are relatively much closer to ones with relative encodings (close to the baseline with different initialisations) than other ones, and that does provide some evidence that it is closer to relative encodings. But it is not clear to me why comparing the attention patterns should help us determine which type of positional information is being used.

---

> ### Author Rebuttal · Authors · 2023-08-09
>
> Thank you, we are delighted that you like our work!
>
> > Novelty of results
>
> Thank you for your insightful comment on the novelty of our results and the framing of our introduction. We agree that the field has seen previous work on Transformers without positional encodings, as you have detailed.
>
> What sets our work apart, however, is the systematic evaluation we conduct on the impact of widely used positional encodings on length generalization specifically in *downstream tasks* for autoregressive (decoder-only) Transformers. The downstream aspect of our work differentiates us with works such [7] and [8]. While other works such as [6] have explored various aspects of Transformers without positional encodings, our focus is unique in its in-depth examination of differences between widely used PEs. Furthermore, our study moves beyond showing that NoPE performs on par with other methods in IID settings, to demonstrate its potential superiority in OOD scenarios, thereby differentiating our findings from [7, 1, 2]. Our theoretical contributions are also distinctive, as we are the first to directly show that both relative and absolute encoding are recoverable in decoder-only transformers, in contrast to the theoretical analysis in [3] (understanding attention through lens of kernels) and [5] (turing-completeness via emulating RNNs).
>
> To ensure clarity for readers unfamiliar with this area of study, we appreciate your suggestion to refine the related work section and the introduction. We will take care to present a more comprehensive review of the prior work and history of NoPE, including the specific contributions you've highlighted, and position our work within this context to reflect its unique contributions to the field. This adjustment will further enhance the paper's accessibility and relevance to its intended audience.
>
> > Effect of scale.
>
> We acknowledge the reviewer’s concerns about the effect of scale. As the reviewers correctly points out, training LLMs in practice requires large amounts of compute budget (even at relatively small scales such as 7B) and expertise, which is only possible at few industrial labs. The challenging nature of such undertaking is evident in the underwhelming performance of various open-source LLMs [9].
>
> With that said, we take initial steps in addressing this concern by providing the preliminary results for scaling the model size and pertaining to the General Response section. These results affirm our initial findings, offering encouragement for their applicability at larger scales. Furthermore, we're actively working on more extensive pretraining experiments with diverse data and larger models (1B), to the extent of our available resources.
>
> > (Q1) Similarity Between Attention Patterns
>
> We appreciate this thought-provoking question, and we recognize the complexity of evaluating positional information through attention patterns. But, we’d like to point out that in our experiments, the models share identical configurations except for the positional encoding, so most variation in their behavior can be attributed to the different PE methods. This provides a foundation for comparing their behavior, although, as you correctly noted, the learning trajectories might diverge across models.
>
> To counteract this potential divergence and offer a more robust measure, we compare all pairs of attention heads within a given layer across multiple seeds. This allows us to observe consistent trends that go beyond individual variations.
>
> It's noteworthy to mention that in the interpretability literature, attention patterns have emerged as a prevalent method to probe the internal mechanisms of Transformer models [10,11]. While this may not provide an absolute understanding of the underlying positional information computation, our findings offer valuable insights into the relative influence of different encodings. The observation that attention patterns of NoPE are closer to relative encodings than others provides preliminary evidence of their internal workings.
>
>
> - [1] DiSAN: Directional Self-Attention Network for RNN/CNN-Free Language Understanding. 2017
> - [2] Assessing the Ability of Self-Attention Networks to Learn Word Order.
> - [3] Transformer Dissection: A Unified Understanding for Transformer's Attention via the Lens of Kernel. 2019
> - [4] On the Ability and Limitations of Transformers to Recognise Formal Languages.
> - [5] On the Computational Power of Transformers and its Implications in Sequence Modeling. 2020
> - [6] Transformers Learn Shortcuts to Automata. 2022
> - [7] Transformer Language Models without Positional Encodings Still Learn Positional Information. 2022
> - [8] Bloom: A parameter open-access multilingual language model. 2022
> - [9] https://kaistai.github.io/FLASK/
> - [10] https://aclanthology.org/P17-1088/
> - [11] https://aclanthology.org/2020.acl-main.312/

---

> > ### Comment · Reviewer_rWd9 · 2023-08-18
> >
> > Thank you for the clarifications. As mentioned, it would be helpful to discuss prior work more comprehensively. Regarding the effect of scale, even though it is quite important, I don't think it is completely fair to expect those experiments due to resource constraints.
> >
> > I had already given a positive score and would like to maintain my score.

---

> > > ### Author Response · Authors · 2023-08-21
> > >
> > > Dear Reviewer,
> > >
> > > We truly appreciate your engagement during the rebuttal and discussion periods, we believe your invaluable input has improved our work. We will make sure to incorporate your feedback in the camera-ready.

---

### Official Review · Reviewer_qZrr · 2023-07-06

**Soundness:** 3 good
**Presentation:** 2 fair
**Contribution:** 4 excellent
**Rating:** 7
**Confidence:** 4

**Summary:**

Length generalization is one of hot topics for transformers. In this paper authors investigate how positional embedding influences this generalization for **decoder-only** models on **reasoning tasks**. Authors compare for variety of tasks (e.g. copy, summation, sorting, etc.) sinusoidal positional embedding, T5, AliBi, Rotary and no positional embedding (NoPE).
First, authors proof by construction that NoPE for decoder-only models is able to learn positional information due to causal masking. Then authors show that none of the positional embeddings generalizes to longer sequences, while NoPE in some cases performs even better than others (though performance is still bad compared to seen lengths). Empirically by comparing attention patterns authors show that NoPE learns attention similar to T5 models. Finally authors also play with scratchpad and show that it is not always helping generalization for longer sequences.

**Strengths:**

- first time (at least I didn't see in prior works) formal proofs that decoder-only models are capable of positional embedding reconstruction (I went through the proof, no issue found with the math)
- emergency for other positional embeddings development or investigation of transformers failure for the longer sequence generalization for reasoning tasks
- interesting empirical results for reasoning tasks showing that mainly all positional embeddings currently widely used do not work for reasoning task on longer sequences while no positional embedding (NoPE) works similarly or even better sometimes
- interesting insights on the NoPE (learnt attention pattern is more similar to models trained with T5 bias) and results on scratchpad

**Weaknesses:**

- overstatement in the test about positional embeddings for the decoder-only models: here it should be clarified about the type of the tasks where we apply, as for LMs we know that e.g. AliBi is working.
- small scale experiments and limited length generalization evaluation.
- no in-depth comparison of the root for the different behaviour e.g. for AliBi between the reasoning tasks and LM task.

**Questions:**

- from the Abstract and introduction I expected to see NoPE working well for GPT-models, while main results are provided for small-scale models and reasoning tasks rather than language modeling itself. I guess could be better to state about reasoning tasks in the abstract in addition to the restriction to the decoder-only models to be considered
- From results (Appendix plots) I can see that actually no any positional embedding (including NoPE) generalized to long sequences for the reasoning tasks, so the statement "Overall, our work suggests that explicit position embeddings are not essential for decoder-only Transformers to generalize well to longer sequences." is not supported as NoPE also doesn't work (works only for very couple cases + sequence length is not very large compared to training one).
- For RoPE did authors try to insert it in every layer similar to T5 or AliBi which are used in every transformer layer? Otherwise could be not fair comparison in the end if the positional embedding saturates over the layers.
- Eq. (2) why do we take min? Also do we consider all possible pairs of attention heads or we do pairing head1 from A and head1 from B? For other part of evaluation I found smart to use another seed with NoPE to measure how it changes to see that it actually like T5.
- line 40-49: even for relative positional embedding it was shown that it could fail to generalize, see "CAPE: Encoding Relative Positions with Continuous Augmented Positional Embeddings", NeurIPS 2021. In the same paper as well as e.g. in "An image is worth 16x16 words: Transformers for image recognition at scale", ICLR 2021 "Conditional Positional Encodings for Vision Transformers", ICLR 2023 there were reported that NoPE can works for the encoder-based models.
- line 301-305: yes, authors consider LLMs style models (which is just decoders) but the data / tasks are different, so that we cannot just generalize from the observations to the LM data as in LM data we saw opposite behaviour. I found confusing the whole paragraph in this context.
- Limitations stated as "not study how large-scale pretraining affects" - AliBi did study on WikiText103 which not large-scale + model was also rather small <1B params. I believe the issue is the data / task type (see below comments).
- Notation of multiplication as dot is weird. Prefer to see normal multiplication in $k.d$ -> $k\cdot d$
- Eq. 29 $W_V$ -> $W_K$
- Fig. E2 for Copy Variant 1: did authors check if bias in T5 is learnt to be negative or positive? As in AliBi it emulate smoothed window attention and far away positions are mainly ignored, while for the task if copy you need to look at the whole sequence indeed. So could be that design of the positional embeddings are not designed for specific tasks where another reasoning on the positions is done (e.g. for LM we just need some summary context, but not exact values for every positions as we don't do e.g. copy task). Wonder to see what T5 bias then learnt to be consistent or disprove this hypothesis. For now the hypothesis seems to be consistent along all plots shown in the Appendix: e.g. for all copy tasks and reverse AliBi doens't work); for sorting Rotary is the best - which can be explained as we can solve sort by just comparing neighbors and reordering them, so strong relative position bias is needed; for other more or less different positional embeddings behave similarly.
- Why do authors consider only 2L sequence generalization and not longer as in AliBi? I see that in the end we are not able to generalize even from 20 positions to 40 positions for simple reasoning tasks, while in LM (e.g. AliBi) they generalize to longer contexts, e.g. 16k tokens. I think in the main text it is overstated that some positional embeddings do not work - there should be clearly expressed "let's consider reasoning tasks different from LM and investigate how there length generalization happens." I find that it would be interesting to have discussion on the difference between LM task and considered in the paper, as right now for me it is not clear why LM generalization with AliBI e.g. works nicely for even 16k tokens while in the simpler reasoning tasks this is not the case. Is it model? data? task itself? I presume we have more complex dependence on the positions and sequence length than in standard LM where simple summarization of far away context could be enough for good generation longer sequences while in reasoning task it is crucially to have proper positions distinguishing.

**Limitations:**

Limitations in the main text, however main restriction on considering non-LM tasks is not clearly delivered in the paper.

**Update: Originally score was 3 for contribution and overall score 6-weak accept. After rebuttal and discussion they are raised to 4 for contribution and 7-accept.**

---

> ### Author Rebuttal · Authors · 2023-08-09
>
> We would like to thank you for your positive and detailed feedback! We now address the key questions and concerns in detail below:
>
> > Clarifying the Scope and Generalization Setup
>
> Thank you for bringing attention to the perceived overstatement concerning generalization capability of our evaluated models. We recognize the importance of clearly defining the scope and context in which our findings apply. Our intention was to explore the impact of positional encoding on length extrapolation in *downstream* tasks, particularly focusing on reasoning and mathematical tasks. We have attempted to articulate this scope in both the **abstract** (lines 4-6, 9-11) and the **introduction** (lines 44-47, 60-61, 63-64), where we describe our evaluation encompassing these domains as downstream tasks.
>
> However, acknowledging your observation, we realize that further clarification may be necessary to prevent any confusion. Therefore, we plan to revise the introduction and abstract to underscore the scope more explicitly. Additionally, we will include a nuanced discussion regarding the distinction between "language modeling perplexity vs downstream performance," as a framework for assessing length generalization.
>
>
> > Small scale & Length Generalization Evaluation
>
> We acknowledge the reviewer’s concern regarding the scale of our models. In the absence of LLMs trained with various PE under the same condition, it is increasingly difficult to evaluate the impact of PEs in pretrained models on length generalization. We'd like to point out that our chosen model size is consistent with approaches taken in recent research [1,2]. Even with these constraints, we ensured a robust evaluation process, conducting comprehensive assessments across numerous tasks and seeds to reinforce our findings.
>
> Moreover, we've already started to address these concerns by including preliminary **larger scales** and **language model pretraining** results in the limited timeframe of the rebuttal period (see General Response). These insights align with our original findings, suggesting they may apply at larger scales. We're gathering resources for pretraining at a 1B-parameter level, albeit on a limited scale, to further enhance the empirical aspect of our work.
>
> > Analysis on discrepancy in ALiBi’s performance in Language Modeling vs Downstream Task
>
> The reviewer makes an astute observation regarding ALiBi inductive bias as “smoothed window attention,” a phenomenon we refer to as Recency Bias in the paper (Sec. 6.1). Indeed, ALiBi (in the its paper) reports successful generalization to longer sequences in terms of language modeling (LM) perplexity (PPL), a contrast to our findings where ALiBi fails in downstream tasks even for much shorter sequences. This is the underlying reason why we opted for 2L, as we identified that models already falter for these shorter sequences.
>
> We hypothesize that this discrepancy stems from the unique combination of the nature of the language modeling task and the recency bias in ALiBi. In general, modeling natural language often requires a shorter context due to human cognitive constraints, and brief information about past context may suffice for predicting the next token. In support of this, Chi et al. [6] have recently shown that ALiBi's length extrapolation on LM PPL can be emulated using windowed attention. (Refer to General Response for more discussion)
>
> However, this inductive bias, while potentially beneficial for LM PPL, does not universally translate to success in downstream reasoning tasks. These tasks often necessitate long-range dependencies, a requirement where ALiBi can underperform. As Reviewer cleverly noted, this is evident in tasks like Copy and Reverse, where ALiBi typically lags behind other relative PEs. Please refer to General Response for much detailed discussion.
>
> > Clarifying the statement in lines 18-20
>
> We acknowledge the concern regarding this sentence. Our aim was to show that explicit PEs underperform NoPE in length generalization, aligning with evidence that they can challenge Transformers [4,5]. We'll revise the wording to accurately convey our findings.
>
> > Inserting RoPE in every layer.
>
> Yes. we apply rotary in every layer, our implementation is borrowed from GPT-NeoX. We will add this detail in the appendix.
>
> > All possible pairs of attention heads and the intuition behind the lower bound on distance in Eq. 2.
>
> Yes, we do consider **all** possible pairs of attention i.e. head_i from A and head_j from B where i, j$ \in$ [1, num_heads]. We chose min to have a more strict evaluation to find if there is any meaningful and significant difference among NoPE and absolute PE encoding models. The evaluation would be more reliable if two models are shown to be different from each other even if we take the min. Nevertheless, taking the mean will also result in the same pattern and ranking (see Fig. 2 in the uploaded PDF)
>
> > References to generalization in relative PEs in vision Transformers
>
> We thank the reviewer for the reference. We will incorporate them in the paper. Also, we'd like to note that in the text domain, encoder-only transformers become bag of words without PE [3].
>
> > Potential Confusion in line 301-305
>
> We appreciate the reviewer's insight. Our intention was to highlight our evaluation's unique focus on decoder-only models and downstream tasks. We will revise this paragraph to eliminate any confusion.
>
> > Typos
>
> Thank you for spotting them! We will correct them.
>
> With these clarifications, would you kindly consider increasing your score?
> - [1] Delétang et al, https://arxiv.org/abs/2207.02098, ICLR 2023
> - [2] Ruoss et al,https://aclanthology.org/2023.acl-short.161.pdf, ACL 2023
> - [3] Haviv et al, https://aclanthology.org/2022.findings-emnlp.99/, EMNLP 2022 Findings
> - [4] Sinha et al, ​​https://arxiv.org/abs/2210.12574, EMNLP 2022 Findings
> - [5] Luo et al, https://aclanthology.org/2021.acl-long.413/, ACL 2021
> - [6] Chi et al. https://arxiv.org/abs/2212.10356, ACL 2023

---

> > ### Comment · Reviewer_qZrr · 2023-08-14
> > **Reviewer response to the rebuttal**
> >
> > Dear authors,
> >
> > Thanks for all the details and clarifications! I have carefully read all reviewers' comments and your responses. Let me add a bit more comments for the sake of community and fruitful discussion we started:
> >
> >  > Clarifying the Scope and Generalization Setup
> >
> > After rereading the abstract and introduction I agree that it is more or less clearly stated in the paper. I would suggest maybe to highlight not only downstream tasks but also reasoning as itself. I think your findings are really very impactful in the direction of reasoning showing that all current solutions are worse than noPE and we did not make any progress even in simple reasoning tasks. Maybe in the discussion you can even add an emergency of new general architectures to be able to solve the problem as it could be that positional embedding is tight to the modeling itself to avoid shortcuts models are doing now.
> >
> > > Small scale & Length Generalization Evaluation
> >
> > I am totally fine with current models size and scale as you highlighted that the purpose is to show that for e.g. simple reasoning tasks any positional embedding doesn't work even for small length increase. So I don't think that absence of large scale is any significant downside of the paper.
> >
> > >  Analysis on discrepancy in ALiBi’s performance in Language Modeling vs Downstream Task
> >
> > You have resolved my concern. What I suggest is to include our discussion into the paper, pointing on the choice of 2L as generalization setting, and also referencing to the works which show issue with AliBi as window attention and discussing reasoning tasks more carefully here (tasks which need to have full attention, e.g. Copy and Reverse). I think this discussion could emerge connection between people in reasoning domain and LLMs domain to collaborate deeper on understanding reasoning for LMs.
> >
> > > Inserting RoPE in every layer.
> >
> > Thanks for the details. I am good with your evaluation here. Thanks!
> >
> > > Evaluation on attention heads
> >
> > I am all good, thanks for the updated figure - good to see consistency here.
> >
> > > References to generalization in relative PEs in vision Transformers
> >
> > Overall across prior works I see visible distinction between encoder, decoder and encoder-decoder archs (not that we should use noPE encoder for text - yes, it is a bag of words which could not produce good embeddings). Would be nice to have some clarification in text on this highlighting that right now we don't have a clear picture for all of them under the same hypothesis. But this is a more philosophical discussion :) .
> >
> > > Fig. E2 for Copy Variant 1: did authors check if bias in T5 is learnt to be negative or positive? As in AliBi it emulates smoothed window attention and far away positions are mainly ignored, while for the task if copy you need to look at the whole sequence indeed. So it could be that the design of the positional embeddings are not designed for specific tasks where another reasoning on the positions is done (e.g. for LM we just need some summary context, but not exact values for every positions as we don't do e.g. copy task). Wonder to see what T5 bias then learnt to be consistent or disprove this hypothesis. For now the hypothesis seems to be consistent along all plots shown in the Appendix: e.g. for all copy tasks and reverse AliBi doens't work); for sorting Rotary is the best - which can be explained as we can solve sort by just comparing neighbors and reordering them, so strong relative position bias is needed; for other more or less different positional embeddings behave similarly.
> >
> > Any comment on my previous question regarding T5? Anything else maybe you see from its values?

---

> > > ### Author Response · Authors · 2023-08-15
> > >
> > > Dear reviewer,
> > >
> > > We appreciate your thorough review of our paper and response. It's gratifying to know our discussion **addressed your queries** related to the “**Scope**”, “**ALiBi performance**”, “**Attention head evaluation**”, and “**Models’ scale**”. We're also heartened that you consider our findings to be “**very impactful**”. We will ensure these discussions are appropriately reflected in the camera-ready.
> > >
> > > > Values of T5 Relative Bias
> > >
> > > Below, we present our analysis on the T5 Relative Bias values in the decoder (in self-attention) of the pretrained HuggingFace model. (If you have a specific task in mind, we'd be happy to revisit this analysis).
> > >
> > > Please refer to Appendix B.2 for a detailed breakdown on T5: Notably, this analysis illustrates the learnt bias value $b_{\mathrm{bucket}}$ for each relative distance $\mathrm{bucket} \in [0, 31]$). (each attention head has a distinct set of such biases). Smaller buckets corresponds to shorter distances between tokens, while larger buckets corresponds to the longer distances. Bucket 31 is exclusively used for all distances $\ge$ 128 tokens.  We observe two primary patterns in the pretrained 3b model (We found these patterns to be consistent across all sizes):
> > >
> > > **Type I** (As demonstrated for head 1 in the below table)
> > >
> > > 1. Decreasing but *positive* values for close buckets: [0, 5]
> > > 2. Decreasing but *negative* values for middle buckets [5, 20], i.e, values are becoming more negative.
> > > 2. Increasing values for buckets [20, 30].
> > > 4. *A significant negative value* for bucket 31 (which corresponds to all relative distances $\ge$ 128 tokens).
> > >
> > > |$\mathrm{bucket}$|Relative Bias Value $b_{\mathrm{bucket}}$|ASCII Representation of Bias Value|
> > > |-|-|-|
> > > |0|2.5469|`-----/#----`|
> > > |1|4.9688|`-----/##---`|
> > > |2|1.6406|`-----/-----`|
> > > |3|0.0967|`-----/-----`|
> > > |4|-0.6484|`-----/-----`|
> > > |5|-1.2422|`-----/-----`|
> > > |6|-1.5703|`-----/-----`|
> > > |7|-1.8516|`-----/-----`|
> > > |8|-2.0469|`----#/-----`|
> > > |9|-2.2344|`----#/-----`|
> > > |10|-2.4062|`----#/-----`|
> > > |11|-2.3906|`----#/-----`|
> > > |12|-2.4062|`----#/-----`|
> > > |13|-2.5156|`----#/-----`|
> > > |14|-2.5938|`----#/-----`|
> > > |15|-2.5625|`----#/-----`|
> > > |16|-2.6094|`----#/-----`|
> > > |17|-2.6719|`----#/-----`|
> > > |18|-2.6562|`----#/-----`|
> > > |19|-2.6094|`----#/-----`|
> > > |20|-2.4688|`----#/-----`|
> > > |21|-2.3750|`----#/-----`|
> > > |22|-2.1719|`----#/-----`|
> > > |23|-1.9297|`-----/-----`|
> > > |24|-1.5938|`-----/-----`|
> > > |25|-1.2812|`-----/-----`|
> > > |26|-0.9141|`-----/-----`|
> > > |27|-0.5508|`-----/-----`|
> > > |28|0.2305|`-----/-----`|
> > > |29|0.5859|`-----/-----`|
> > > |30|0.9805|`-----/-----`|
> > > |31|**-27.7500**|`#####/-----`|
> > >
> > > **Type II** (As demonstrated for head 2 in the below table)
> > >
> > > The same pattern is observed across buckets [0, 30], but, in contrary, we observe *very large positive value* for bucket 31.
> > >
> > > |$\mathrm{bucket}$|Relative Bias Value $b_{\mathrm{bucket}}$|ASCII Representation of Bias Value|
> > > |-|-|-|
> > > | 0 | 2.5000 | `-----/#----` |
> > > | 1 | 2.5469 | `-----/#----` |
> > > | 2 | 1.5234 | `-----/-----` |
> > > | 3 | 0.7383 | `-----/-----` |
> > > | 4 | 0.1758 | `-----/-----` |
> > > | 5 | -0.2832 | `-----/-----` |
> > > | 6 | -0.8281 | `-----/-----` |
> > > | 7 | -1.1719 | `-----/-----` |
> > > | 8 | -1.4375 | `-----/-----` |
> > > | 9 | -1.8516 | `----#/-----` |
> > > | 10 | -1.9219 | `----#/-----` |
> > > | 11 | -2.1875 | `----#/-----` |
> > > | 12 | -2.3438 | `----#/-----` |
> > > | 13 | -2.2656 | `----#/-----` |
> > > | 14 | -2.4062 | `----#/-----` |
> > > | 15 | -2.5625 | `----#/-----` |
> > > | 16 | -2.5781 | `----#/-----` |
> > > | 17 | -2.6406 | `----#/-----` |
> > > | 18 | -2.7500 | `----#/-----` |
> > > | 19 | -2.8125 | `----#/-----` |
> > > | 20 | -2.7500 | `----#/-----` |
> > > | 21 | -2.5469 | `----#/-----` |
> > > | 22 | -2.4375 | `----#/-----` |
> > > | 23 | -2.2969 | `----#/-----` |
> > > | 24 | -2.0000 | `----#/-----` |
> > > | 25 | -1.6719 | `----#/-----` |
> > > | 26 | -1.4375 | `-----/-----` |
> > > | 27 | -0.9844 | `-----/-----` |
> > > | 28 | -0.2734 | `-----/-----` |
> > > | 29 | 0.0588 | `-----/-----` |
> > > | 30 | 0.5312 | `-----/-----` |
> > > | 31 | **7.7812** | `-----/#####` |
> > >
> > >
> > > Diving deeper into **bucket 31**, its value across all heads (coincidentally, this model has 32 heads) is:
> > >
> > > |Head|Relative Bias Value for bucket 31 ($b_{31}$)|
> > > |-|-|
> > > |1|-27.75|
> > > |2|7.78125|
> > > |3|-28.75|
> > > |4|-29.125|
> > > |5|-31.375|
> > > |6|-32.75|
> > > |7|28.0|
> > > |8|12.4375|
> > > |9|33.5|
> > > |10|-25.875|
> > > |11|-35.25|
> > > |12|26.25|
> > > |13|28.75|
> > > |14|-27.875|
> > > |15|-39.5|
> > > |16|-26.375|
> > > |17|69.5|
> > > |18|-30.125|
> > > |19|-34.75|
> > > |20|-26.5|
> > > |21|-30.375|
> > > |22|45.75|
> > > |23|-32.5|
> > > |24|38.75|
> > > |25|-28.0|
> > > |26|29.0|
> > > |27|16.5|
> > > |28|4.75|
> > > |29|-31.0|
> > > |30|43.0|
> > > |31|-31.875|
> > > |32|-22.5|
> > >
> > > Mean = -5.883, SD = 31.349
> > >
> > > Our observations suggest that while T5 has heads that mask distant tokens (Type I), other heads attend to distant tokens when required (Type II). This nuanced distinction could potentially account for the performance differences between T5 and ALiBi, as T5 can learn to adaptively attend to distant tokens, in contrast to ALiBi, which always (yet smoothly) mask distant tokens.
> > >
> > > We thank you again for your time and feedback and hope our response answers your inquiry. Are there any remaining questions you would like us to answer?

---

> > > > ### Author Response · Authors · 2023-08-18
> > > >
> > > > Dear Reviewer,
> > > >
> > > > Thank you for your invaluable feedback and the insightful discussion initiated. We're curious to know if you've had a chance to go over our previous response. We hope it has addressed your question. As the discussion phase nears its end, we're eager to address any outstanding questions you may have, and to incorporate any additional feedback. We hope you kindly consider a renewed assessment of the paper, considering our recent exchanges. Once again, we appreciate your time and dedication to the review process.

---

> > > > > ### Comment · Reviewer_qZrr · 2023-08-19
> > > > > **Final comments and raising the score**
> > > > >
> > > > > Dear authors,
> > > > >
> > > > > Sorry for delay (busy week). I had a look at your analysis of T5 - this is amazing! I found it is very interesting that T5 is learning some heads to ignore distant tokens and some in contrary. Also this shows that something non-trivial happens with distant tokens for embeddings which do not do window attention. I am happy with the analysis you provided and strongly recommend (and hope) you to include it into Appendix for the final revision.
> > > > >
> > > > > As I said before I think the paper provides very important analysis and results to understand better transformers, reasoning tasks and generalization. I am raising my score.
> > > > >
> > > > > Thanks again for nice and productive discussion!

---

> > > > > > ### Author Response · Authors · 2023-08-21
> > > > > >
> > > > > > Dear Reviewer,
> > > > > >
> > > > > > We're truly grateful for your active participation during the rebuttal and discussion phases. Your feedback has been instrumental in enhancing the quality of our work. We will incorporate the changes promised in the final version.

---

### Official Review · Reviewer_VbrS · 2023-07-10

**Soundness:** 3 good
**Presentation:** 3 good
**Contribution:** 2 fair
**Rating:** 5
**Confidence:** 3

**Summary:**

The paper presents a comprehensive study on the role of positional encodings in Transformer models, focusing on their ability to generalize over sequence lengths. The authors introduce a novel positional encoding method, which does not use explicit positional information.

**Strengths:**

1. The paper introduces a novel positional encoding method, which is a significant contribution to the field.
2. The paper is well-written and easy to follow.

**Weaknesses:**

 The authors did not study how large-scale pretraining affects different PEs due to the lack of publicly available large language models trained with various PEs under similar conditions. This leaves a significant aspect of the problem unexplored.

**Questions:**

1. How would NoPE perform in a broader range of tasks beyond the mathematical and reasoning tasks evaluated in this study?
2. Could the authors elaborate on the potential implications of their findings for the design of future Transformer models?

---

> ### Author Rebuttal · Authors · 2023-08-09
>
> We thank the reviewer for their time in providing useful feedback and questions which we now address:
>
> > Effect of large-scale pretraining on PEs
>
> Thank you for raising the concern regarding the exploration of large-scale pretraining effects on different PEs. It's worth noting that the pretraining of large language models (LLMs) demands significant computational resources. For instance, training StartCoder-13B necessitated ~320K hours on NVIDIA A100 80G GPUs [1], while MPT-7B utilized a cluster of about 500 A100 GPUs for 10 days, translating to a cost nearing $200K [2]. Given the constraints of our available resources, such extensive undertakings are well beyond our capacity. Although the scale of our models was limited, we made sure to perform extensive evaluations across many tasks and seeds to support our conclusions.
>
> Having said that, we have provided preliminary results on larger-scale models and pretraining experiments in the General Response. These scaled up experiments exhibit patterns *consistent with those presented in our submission* suggesting the same result might hold for larger scales. Moving forward, we are preparing the necessary resources for pretraining at 1B parameters (albeit at limited scale) to further improve the empirical results of the draft.
>
>
> > NoPE performance in a broader range of tasks
>
> We would like to point out that although we used mathematical and reasoning tasks—common benchmarks in length generalization literature [3,4]— in our evaluation, these are foundational and can be seen as representative of more complex tasks. Nonetheless, our empirical investigation suggests that NoPE overfits less than other explicit PEs, making it a suitable choice in scenarios prone to positional shortcuts [6].
>
> Moreover, results in section 6.1 demonstrate that NoPE, in contrast to ALiBi, can reliably attend to both long and short range dependencies. Coincidentally, Liu et al [5] recently observed that LLMs are more sensitive to context provided early or late in the context. This suggests that LLMs trained with NoPE could potentially offer more benefits in such contexts when compared to those trained with ALiBi.
>
>
> > Implications of our findings
>
> We believe our findings can impact the development of future Transformer models in two primary ways:
>
> First, as acknowledged by Reviewer 2 (qZrr), our finding on the commonly used PEs (such as Rotary & ALiBi) can inform the community about their potential shortcoming especially in terms of length generalization and encourage further development and research on better positional encoding methods. This is especially important as training LLMs is an extremely expensive and resource-intensive undertaking; any hidden shortcomings in the model architecture could be vastly wasteful and harmful to the environment.
>
> Secondly, and more notably, our theoretical and empirical analyses reveal that decoder-only Transformers possess inherent mechanisms for representing positions. These mechanisms perform better than or on par with other explicit PE schemes. This, as suggested by Reviewer 4 (C4oP), could guide the design of more streamlined and simpler Transformer architecture and positional encodings that leverage and strengthen these internal mechanisms.
>
> We hope that the clarifications provided have resolved any lingering questions. Would you consider increasing your ratings  given the main clarifying points outlined?
>
> - [1] Li et al, StarCoder: may the source be with you!, 2023
> - [2] https://www.mosaicml.com/blog/mpt-7b, 2023
> - [3] Delétang et al, https://arxiv.org/abs/2207.02098, ICLR 2023
> - [4] Ruoss et al,https://aclanthology.org/2023.acl-short.161.pdf, ACL 2023
> - [5] Liu et al., https://arxiv.org/pdf/2307.03172.pdf, 2023
> - [6] Sinha et al, ​​https://arxiv.org/abs/2210.12574, EMNLP 2022 Findings

---

> > ### Author Response · Authors · 2023-08-18
> >
> > Dear Reviewer,
> >
> > Thank you again for your time for reviewing our paper. We would appreciate knowing if you were able to look at the response we provided. With the discussion period nearly over, we would love to answer any remaining questions you may have.

---

### Author Rebuttal · Authors · 2023-08-09

## General Response
We thank the reviewers for their positive feedback and constructive criticism.

We are glad that Reviewers 1 (VbRS) and 3 (rWd9) found our paper to be **“well-written and easy to follow,”** and that they believe it provides a “thorough and in-depth analysis,” as highlighted by Reviewer 4 (C4oP).

We are further pleased that Reviewer 2 (qZrr) acknowledges the **mathematical rigor** in our work, and Reviewer 3 (rWd9) recognizes our experimental rigor, especially noting that our evaluation methodology is “**more aligned with practice** than prior works that have explored this direction.”

Reviewer 2 (qZrr) kindly underscored the **originality** of our formal proofs and insights such as the "learned attention patterns of NoPE" and its working mechanism, also noted by Reviewer 4 (C4oP). We further appreciate Reviewers 3 (rWd9), 2 (qZrr), and 4 (C4oP) recognizing our interesting and surprising results, strengthening our confidence in the innovative nature of our findings.

Finally, we are glad Reviewers 1 (VbrS) and 3 (rWd9) considered our work **"a significant contribution"** and "a step forward in understanding Transformers with causal masking", reflecting our aim to address this important aspect. Moreover, we appreciate Reviewers 2 (qZrr) and 4 (C4oP) recognizing our impact on development of positional encoding and more simplified Transformer architectures.

We now clarify the main shared concern among the reviewers below and address reviewer specific questions in the individual responses.

### Language Modeling and Larger Models
During the limited time of the rebuttal, we performed preliminary **pretraining** experiments on **larger scale** models. Specifically, we trained a 300M decoder-only Transformer with NoPE and ALiBi on Wikitext103 with training context size of *512* (following Press et al [1]).

We observe almost identical perplexity for both NoPE and ALiBi on the test set with context sizes of 128 to 630. From context sizes of 630 to 1120, we see that NoPE's perplexity diverges but ALiBi's perplexity stays the same. However, when we take these exact checkpoints (that are pretrained on Wikitext103) and finetune them on downstream tasks, we observe NoPE still outperforms ALiBi (Figure 1 in the uploaded PDF), which is consistent with results reported in the original draft.

**Language Modeling Perplexity vs. Downstream Performance**

As we argue in the manuscript, we believe that language modeling perplexity is not well suited for evaluating length generalization since the downstream task performance is what we really care about in practice: precisely the focus of our evaluation.There are various observations in the literature showing that perplexity is not correlated [2,3] and sometimes is anti-correlated [4] with downstream task performance.

More importantly, Chi el al. [5] recently showed that ALiBi can be emulated using window attention. That is, NoPE+windowed attention mask (where every token attends only to those within $w$ tokens of itself, and $w$ is the window size $\lt$ context size) can achieve length generalization in term language modeling perplexity similar to ALiBi [5]. Furthermore, Chi el al. [5] demonstrate that large slope magnitude in ALiBi formulation is needed for length generalization as large slopes inherently create a narrow window attention mask such that far away tokens are ignored.

These results suggest that Recency bias incorporated in ALiBi's design (as suggested by Reviewer 2 (qZrr)) might create an unrealistic impression of how well it generalizes over length in natural language modeling. Especially, as the prediction of the next token may only need a brief summary of the past context and natural language inherently contains local dependencies due cognitive constraints [6, 7]. On the other hand, for complex reasoning tasks that require long term dependencies, the Recency Bias of ALiBi can hinder the model, while positional encoding such as NoPE can freely attend to them (Fig. 7 in the main draft). The limitation of Recency Bias in ALiBi can be observed in the Reverse and Copy tasks (Fig. 3 in the main draft) where long range dependencies are needed.

We understand that these details may not have been sufficiently discussed in the original draft but we hope these explanations fully address the questions by reviewers. We make sure to have a more comprehensive discussion of LM vs. Downstream in the updated manuscript.

Nevertheless, we are preparing our resources (to the best of our ability) to train larger scale (1B) models with NoPE and ALiBi to fully reflect the effect of pretraining. We will update the camera-ready with the new results.

- [1] Press et al, Train Short, Test Long: Attention with Linear Biases Enables Input Length Extrapolation, ICLR 2022
- [2] Tay et al. Scale Efficiently: Insights from Pretraining and Finetuning Transformers, ICLR 2022
- [3] Liu et al. Same Pre-training Loss, Better Downstream: Implicit Bias Matters for Language Models, 2022
- [4] Zhou et al. LIMA: Less Is More for Alignment, 2023
- [5] Chi et al. Dissecting Transformer Length Extrapolation via The Lens of Receptive Field Analysis, ACL 2023
- [6] Gibson et al, Linguistic complexity: locality of syntactic dependencies, 1998
- [7] Gibson et al, How efficiency shapes human language, 2019

---

### Decision · Program_Chairs · 2023-09-21

**Decision:**

Accept (poster)

**Comment:**

The paper examines the extrapolation behavior of transformers without positional embeddings, showing strong results on several algorithmic tasks.

Reviewers appreciated the well motivated experiments, interesting empirical results, and theoretical justification. There is also interesting analysis, such as the comparison with T5 positional embeddings.

There are some concerns about overclaiming. The title/abstract should do a better job of discussing the scope of tasks they experiment with (for example, some reviewers were disappointed by the lack of natural language tasks such as summarization). Some reviewers were also left with the impression that NoPE is novel, so the paper needs to acknowledge related work more clearly. While the paper doesn't say anything false here, I think the presentation is somewhat misleading.

The paper would also benefit from discussing behavior on pre-training tasks. The author response mentions an interesting experiment showing that NoPE *doesn't* extrapolate well for language modeling - including this result, and discussing the reasons for different conclusions to algorithmic tasks, would be very interesting (at least to me).

Overall the paper is borderline, but leaning accept.